# Drought Research Priorities, Trends and Geographic Patterns

Roland Baatz[1,*], Gohar Ghazaryan[1,2], Michael Hagenlocher[3], Claas Nendel[1,4], Andrea Toreti[5] and Ehsan Eyshi Rezaei[1]

[1] Leibniz Centre for Agricultural Landscape Research (ZALF), Müncheberg, Germany
[2] Geography Department, Humboldt-Universität zu Berlin , Unter den Linden 6, 10099 Berlin, Germany
[3] United Nations University, Institute for Environment and Human Security (UNU-EHS), Platz der Vereinten Nationen 1, 53113 Bonn, Germany
[4] Institute of Biochemistry and Biology, University of Potsdam, Germany
[5] European Commission, Joint Research Centre, Ispra, Italy

*Correspondence to*: Roland Baatz roland.baatz@zalf.de

**Abstract.** Drought research addresses one of the major natural hazards that threatens progress toward the Sustainable Development Goals. This study aims to map the evolution and interdisciplinarity of drought research over time and across regions, offering insights for decision-makers, researchers, and funding agencies. By analyzing more than 130,000 peer-reviewed articles indexed in Scopus from 1901 to 2022 using Latent Dirichlet Allocation (LDA) for topic modeling, we identified distinct shifts in research priorities and emerging trends. The results reveal that plant genetic research for drought-tolerant genotypes and advancements in drought forecasting are the most dominant and continuously growing areas of focus. In contrast, the relative importance of topics such as ecology, water resource management, and forest research has decreased. Geospatial patterns highlight a universal focus on forecasting methods, with a strong secondary emphasis on policy and societal issues in Africa and Oceania. Interdisciplinarity in drought research experienced a marked decline until 1983, followed by a steady increase from 2007 onward, suggesting a growing integration of diverse fields. Emerging topics in recent years signal evolving priorities for future research. This analysis provides a comprehensive overview of drought research trends across sectors and regions, offering strategic guidance for aligning research efforts with drought resilience goals. The findings are crucial for research funding agencies and policymakers aiming to prioritize areas with the highest potential to mitigate drought impacts effectively.

## 1    Introduction

Drought is one of the socio-economically most damaging natural hazards (Yin et al., 2023; Esha Zaveri et al., 2023). Contrary to other climate extremes, drought manifests on a vast spatiotemporal scale, extending up to thousands of kilometers, and can persist for periods lasting up to years (Mondal et al., 2023). Drought episodes are becoming increasingly frequent, extreme,

and prolonged driven by climate change (Hoylman et al., 2022; IPCC, 2021). Drought is significantly tied with other climate-driven hazards, particularly heatwaves, which can amplify drought impacts (Lesk et al., 2022). The emergence of frequent flash droughts in over 74% of the globe during the last 64 years has been also recently revealed (Yuan et al., 2023). This pattern is largely linked to elevated anomalies in evapotranspiration and precipitation deficits which are confirmed consequences of human-induced climate change (Yuan et al., 2023). Drought as the state of water shortage is exacerbated by anthropogenic activities such as unsustainable water use, allocation and water extraction (van Loon et al., 2022; van Loon et al., 2016a; Chiang et al., 2021). This led to the reconsideration of the definition of drought as rather a being a process than a system state (van Loon et al., 2016a; AghaKouchak et al., 2021; van Loon et al., 2024). Drought poses therefore a substantial risk for and across sectors and systems (Voosen, 2020; Walker and van Loon, 2023; Hagenlocher et al., 2023), including agriculture, water supply, health, the energy sector, ecosystem services, and socio-political stability.

Drought impacts ecosystems by modifying ecological processes, altering of community structures and composition (Canarini et al., 2021). These changes can lead to adaptations such as improved water use efficiency in response to water storage(Poppe Terán et al., 2023). The total land area and population affected by severe terrestrial water storage drought could more than double by the end of the twenty-first century (Pokhrel et al., 2021). Over the first twenty years of the 21st century, extreme drought and drinking water shortages have plagued more than 80 major cities worldwide (Savelli et al., 2023). Food production and security have already been largely compromised by drought (Spinoni et al., 2020; Rossi L et al., 2023). For instance, the size of the dry zones across the global grain production area increased by 1.1 % per decade in the period from 1951 to 2011 (Wang et al., 2018). Globally, the average national cereal production shrank by 10 % over the period 1964 to 2007 as a result of extreme drought and heat (Lesk et al., 2016). In Europe, the adverse impacts of droughts and heatwaves on crop production tripled in the last 50 years (Brás et al., 2021).

Without climate action, annual drought damages for the EU and the UK could escalate from €9 billion to over €65 billion per year by 2100, doubling in terms of financial impact (Naumann et al., 2021). Drought also impacts human health (Vins et al., 2015) e.g. through reducing stream flow, increasing concentration of pathogens, enabling some vector-borne diseases (Cann et al., 2013) and as risk factor of child undernutrition in particular in low-income conditions (Belesova et al., 2019). The impacts of drought are scale specific, event specific and often difficult to quantify due to their indirect and often systemic character - affacting not only human health and agriculture but also energy and social systems (van Loon et al., 2019; Blauhut et al., 2015).

This study is motivated by the need to enhance our understanding of the evolving landscape of drought research, particularly in light of the escalating challenges posed by climate change and water scarcity. While previous reviews outlined the need to synthesize the immense body of literature on drought research (Stein et al., 2022), our analysis distinguishes itself through the use of a data-driven, unsupervised machine learning approach to examine over 130,000 peer-reviewed articles. By exploring

long-term research trends, we identify critical shifts in thematic focus, fundamental and emerging trends, and interdisciplinary collaboration opportunities that have shaped the field. This unique approach allows us to reveal previously overlooked patterns and gaps in the literature, offering insights into how research priorities have been set by the global research communities. Our findings contribute to the development of more effective and systemic drought resilience frameworks by quantifying the connections between diverse research topics, ultimately guiding more strategic alignment of efforts among scientists, funding bodies, and policymakers.

## 2    Methods

### 2.1    Data

We based the analysis on 131,748 abstracts curated in the licensed Scopus database under the search term drought on March 22, 2023. Data on title, keywords, language, abstract and publication year were retrieved from the Scopus database via the Scopus Search API and the elsapy search library. We removed duplicates, copyright information and non-English abstracts. Scopus provides the a curated database of scientific literature and grants access to data- and text mining to licensed users for academic purpose. The following alternative large databases for meta-information of scientific literature were considered: OpenAlex, Web of Science, Dimensions and Semantic Scholars. We chose Scopus because of its high quality of information and granted access for research purpose.

### 2.2    Topic Modelling by Unsupervised Machine Learning

To discern pertinent topics and subtopics within the dataset, we used the Latent Dirichlet Allocation (LDA) (Blei et al., 2003; Radim Rehurek and Petr Sojka, 2010) method. LDA, an iterative Bayesian method of unsupervised machine learning, identifies multiple topic clusters within documents based on keyword distribution, co-occurrence, and frequency. Depending on the chosen granularity, classification can yield either broad or highly-specific topics. Remarkably, while LDA is an established method (Eker et al., 2018; Ewert et al., 2023; Cebral-Loureda et al., 2023; Rahman et al., 2022; Callaghan et al., 2020), its application to vast scientific abstract corpora is rare. Compared to other alternatives, LDA allows for multiple topics within a single document. Also, LDA represents a compromise between computationally more expensive and more costly topic modelling approachs such as BERTopic (Ogunleye et al., 2023), and simpler and computationally less expensive approaches such as Latent Semantic Analysis (Deerwester et al., 1990). To explore the drought research areas, we identified rather general topics and mores specific topics. This was done by pre-defining the number of topics to the algorithm. We then calculated topic distributions with LDA for the documents and for given number of topics based on overall and document specific keyword distributions. We assessed coherence scores for a consecutively increasing number of topics, found that coherence increases until fifty topics, and decided to cap granularity at fifty topics, which would still yield 2634 documents on average per topic. We then selected five topics as a reasonable number for the general classification level, twelve topics for a median

level of granularity and fifty topics for the finest level granularity (see results Section Figure 2). Naming conventions for topics
were derived from pivotal keywords within the context of drought research. To evaluate the evolving significance of research themes over time, we charted relative shares of each topic annually.

### 2.3 Data Post-Processing

For topic congruence, we calculated the cosine similarity between topic pairs within each individual document. A high similarity score indicates that two topics appear more frequently together in the same document. A low similarity score
indicates lesser joint appearance. Cosine similarity normalizes the similarity score by the overall share of the two topics. This allows for a better direct comparison for topics with high shares and those with low shares. Topical overall similarity index is calculated as mean cosine similarity of a topic for the other n-1 topics. Heat maps of cosine similarity are ranked by overall similarity score of a topic starting with the highest to lowest. We visualized the topic trajectory using a Sankey diagram to highlight how general topics with coarse granularity narrow down to more specific topics. Consistent with Sankey diagrams,
the width of the connecting lines is proportional to the document counts they represent. The geographic reference of drought research to individual continents was identified by keyword search. Abstracts were associated to a region i.e. continent if a specific continent, or country (for the US also the states) was mentioned in the title, keyword or abstract. One document could be associated with several regions in case several mentions. Topical shares by region were then calculated based on the documents found.

### 3 Results

### 3.1 Major and specific topics in drought research

The number of drought-related peer-reviewed publications has increased exponentially adding 12,338 articles in 2022 alone (Figure 1). The proportion of articles focussing on drought increases year by year compared to the general scientific literature.
This is expressed by the ratio of drought related research compared to the available scientific publications in webofscience (Figure 1). We let the LDA identify five major topics across the document pool (tier 1), twelve more focused yet still rather general topics (tier 2) and fifty even more specific topics (tier 3, Figure 2). While keyword frequency and co-occurrence generate topic clusters, they also allow domain experts to name the topics according to their context. A list of publications with the highest share of each topic (i.e. topical relevance > 98%) confirms the topic naming.

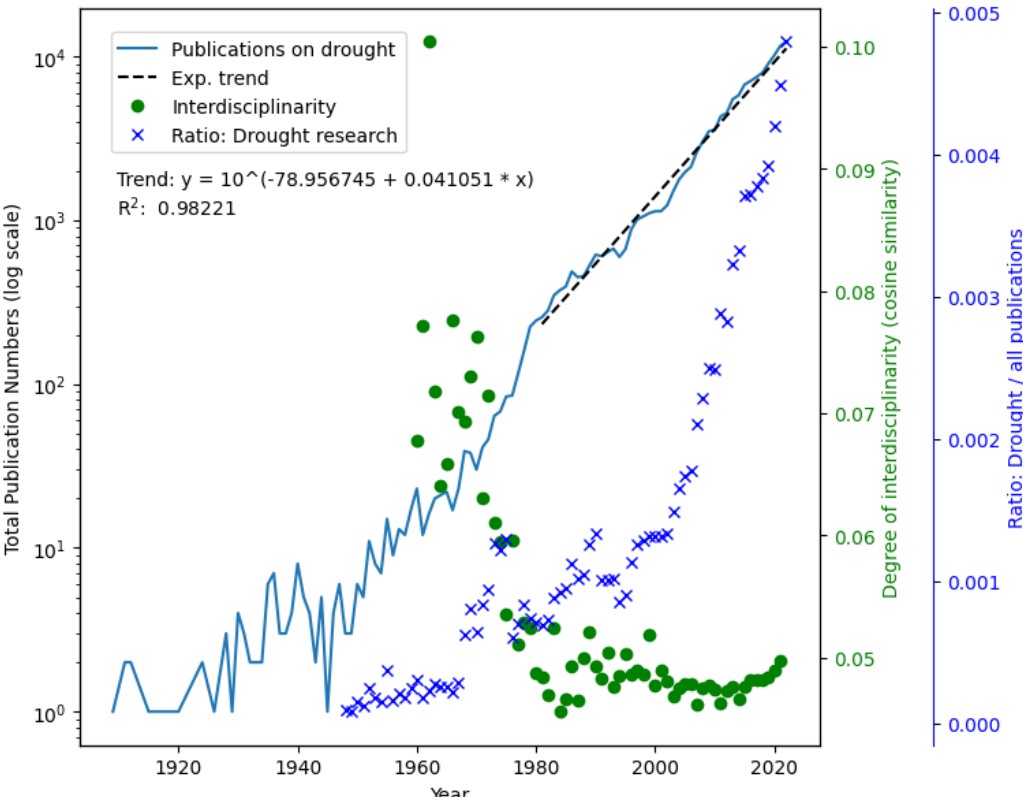

**Figure 1: Publications by year in drought research. Research abstracts listed in Scopus and analyzed over the past century with regard to interdisciplinarity. Drought research exhibits an exponential trend (R2 = 0.98). This trend is highlighted by the increasing ratio of drought research to overall research publications. Interdisciplinarity is calculated as cosine similarity index which is the normalized cross-topic intersection within a document. Focus on specific topics increased until 1980s which is marked by a decline**

**in interdisciplinarity. 1980 onwards plant genetics took a rise, leading to ups and downs in interdisciplinarity. From 2007 onwards inter-disciplinary rose again consistently.**

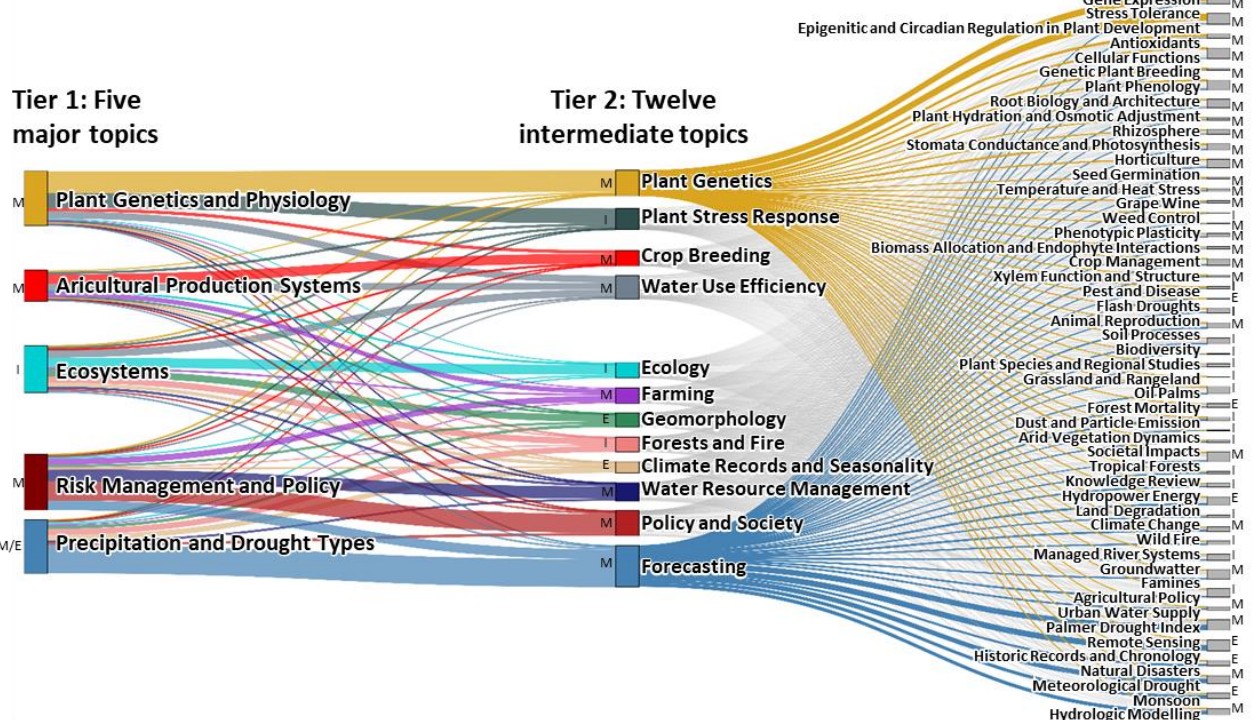

**Figure 2: Hierarchical Depth of Research Topics in Drought Research. The breadth of drought research encompasses a diverse array of subjects. Each column represents the entire research corpus, encompassing 131,748 articles. This figure displays the share of each topic against the whole corpus. Three levels of specificity are distinguished: five broad topics (left), twelve intermediate topics (center), and fifty highly specialized topics (right). Visual emphasis is placed on the flows of plant genetics and forecasting, given their upward trends and dominance in the research field. Line width and width of bars is proportional to the share per topic. Topics are categorized into methods and processes (M), events and historical analysis (E), and impacts on socio-ecosystem compartments (I). For high resolution interactive Figure, please see the Supplement Figure S1.**

The five general topics were categorized as *plant genetics and physiology*, *agricultural production systems*, *ecosystems*, *risk management and policy*, and *precipitation and drought types*. At the medium granularity with twelve topic classification, we identified *forecasting* methods including drought types and events on the one end and *plant genetics* at the other end as most dominant research topics. For the twelve topic classification, a network graph visualizes the connections of topics and keywords (Figure 3). Here, the dataset's structure is visualized with limited number of most important keywords in two dimensional space. As a result of the LDA, and topical distribution, *forecasting* is mainly associated to *risk management and policy* and *precipitation and drought types* from tier 1, with lesser association to the other three topics of tier 1 (Figure 2). This reflects how *forecasting* focuses on answering questions on risk using quantitative methods. In contrast, *water use efficiency* is strongly associated with three topics of tier 1, reflecting a higher transdisciplinarity and positioning it at the intersection of

natural *ecosystems*, *agriculture* and *plant physiology*. *Farming*, as tier 2 topic, is particularly interesting as it bridges the gap through its strong association to *risk* and *agricultural production systems* (Figure 2). Other topics at level two are *geomorphology*, *forests and fire*, *climate records and seasonality*, *water resource management*, and *policy and society* (Figure 3).

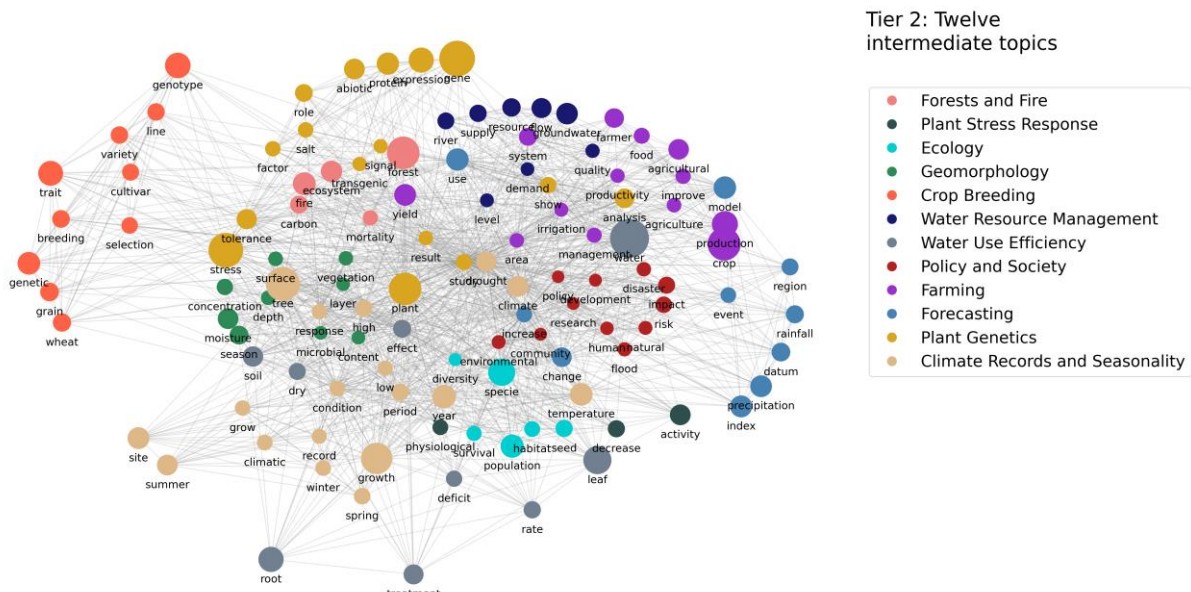

**Figure 3: Network graph of drought research.** Visualizes the network of the twelve drought research topics. Colors denote topics with strongest association, bubble size identifies abundance of keywords, keywords are located below the bubbles, links between keywords identify strongest co-occurrences, and keywords are located in order to minimize connection lengths. For high resolution Figure, please see the Supplement Figure S2.

Topic identification and naming at the finest granularity reveal 50 specialized, potentially emerging topics, with an average of 2634 papers per topic. For these fifty topics, shares by topic ranged between 134 documents and 5901 documents with a median of 2337 documents. In the Sankey diagram (Figure 2), the association of Tier 3 topics with Tier 2 topics is indicated by a) their vertical alignment and b) the strength of the links. The topics *gene expression* and *stress tolerance* are at the top, with a strong link to *plant genetics*. At the bottom of Figure 2, the topics m*onsoon* and *hydrologic modelling* have a strong link to *forecasting*.

These results quantify the extent, importance and the role of research topics with respect to drought research. The results demonstrate the importance to render and limit the topical scopes of reviews.

Other categorizations and classifications are possible. For example, research can be grouped into publications focussing on the analysis of methods and processes, of events and historical analysis, and impacts on socio-ecosystem compartments. Method related research focuses on processes, tools, and methods for adaptation and mitigation of drought. Event studies focus on specific event types and historical analysis to guide development of methods. Topics focussing on impact mostly focus on specific socio-ecosystem compartments. Following this categorization, tier 3 topics with a strong link to *forecasting* can be grouped for example into methods related topics (e.g. *remote sensing*, *hydrologic modelling, meteorological drought*) and events related topics (e.g. *historic drought records and chronology*, *monsoon* (Figure 2)). The density of method-related topics is much higher in the areas of *plant genetics*, *physiology*, and *agricultural management* than in forecasting-related topics. Impact-related topics for tier 3 are mostly found in the center, while topics related to events are predominantly situated in the lower section. This categorization was assessed for tier 1 and tier 2 topics (Figure 2). The results highlight a significant interconnection between research on events, impacts, and methods. This interplay, however, sometimes leads to challenges in distinctly categorizing topics, as evidenced by occasional overlaps and blurred boundaries among these categories.

## 3.2     General and emerging trends

Interest in research topics fluctuates over time. Shifts in research priorities are influenced by societal interest and advancements in technological capabilities. We explored the development over time for drought-related research topics and their relative contributions over the past four decades (Figure 4) and more recently, referring to the years 2012-2022. We chose the last four decades because the data showed a rather high variation in relative contributions for the year before 1982. *Plant genetics* and *forecasting* as well as *crop breeding* are getting an increasing relative share of the research over the last four decades (Figure 4 and Appendix Figure B). *Ecology* and *water use efficiency* have received comparatively less attention with declining shares. Surprisingly, *water use efficiency* is not the specific aim of *plant genetics* and *crop breeding* efforts but rather stands next to these in tier 2. Crop scientists have long targeted drought-tolerant crops to tackle food production challenges in dry regions. The introduction of new genomic technologies has greatly enhanced this effort (Anders et al., 2021), a trend reflected in our analysis. Amongst *plant genetic* and *plant breeding*, specific research on cellular and molecular functions exhibit positive trends for recent years such as the *epigenetic and circadian regulation in plant development* or the role of *antioxidants* that may reduce oxidative damage during drought stress (Bailey-Serres et al., 2019). Plant physiological processes such as *plant phenology* and *stomata conductance* and *photosynthesis* recently became less relevant although strongly related to *plant genetics* and *plant breeding*.

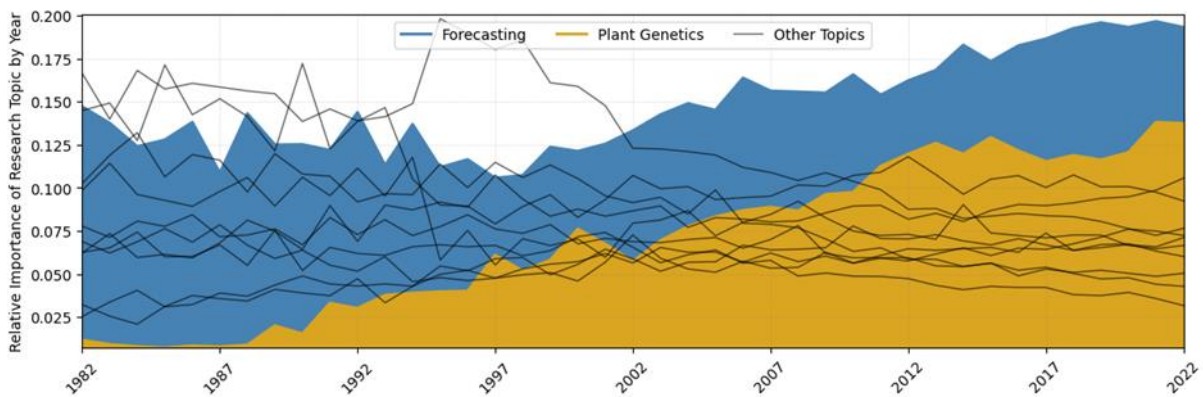

**Figure 4: Temporal Dynamics of Research Topics. The evolution of research topics and their proportionate dominance from 1982 onwards, highlighting the ascendance of plant genetics and forecasting as dominant trending topics. For high resolution Figure, please see the Supplement Figure S3.**


Other tier 2 topics, such as *water resource management*, *geomorphology*, *policy and society*, have also demonstrated a generally declining relative contribution to drought research over the last 40 years (Figure 4 and Appendix Figure B). It is vital to clarify that the decrease in relative share for e.g. *ecology* or *water resource management* topics does not imply a decrease in the absolute number of research studies on these subjects. Instead, the number of documents on these topics continues to

grow, but the rate of growth was comparatively slower (Appendix Figure A). In tier 2, *forecasting* methods and events represented the largest contribution among all topics across the study period (Figure 3).

Looking at Tier 3 topics, we identified the top 10 emerging topics. *Meteorological drought*, *remote sensing*, *climate change*, *natural disasters* and *palmer drought index* are the five emerging topics with strong association to *forecasting*. *Antioxidants*,

*Epigenetic and circadian regulation in plant development*, and *gene expression* are the three emerging topics with strong link to *plant genetics*. *Agricultural policy* and *rhizosphere* are also amongst the ten emerging topics. Here, *agricultural policy* is strongly associated to *policy and society*. Interestingly, *rhizosphere* is the only topic of these with rather evenly strong links to many of the Tier 2 topics.

**3.3    Interdisciplinarity of drought research**

Each document consists of a variety of topics expressed as percentage i.e. share with the total sum of one for each document. While the algorithm for topic identification aims to discern individual topics within the corpus, the major share of a specific topic may dominate a specific document. In another case the topical shares may be dispersed across many topics. We measure interdisciplinarity by cosine similarity, a measure for similarity between two topics and a measure that scales well with the

size of the two topics. We find for tier 2 topics (Figure 5) several robust thematic overlaps. For example, *Plant stress response*

and *water use efficiency* showed the highest thematic overlap. Also, *climate records and seasonality* are strongly manifested in sedimentary records and tree ring records, causing a high similarity with *forests and fire*. Pronounced similarity is also found between *policy and society* with the topics *farming* and *water resource management* (Figure 5). Here, *water resource management* which is crucial for fresh water supply and energy systems (Jasechko et al., 2024) as well as for irrigation and
food security has strong impact and link to *policy and society* (Figure 5).

In contrast, forecasting is rather focusing on short-term responses with less pronounced similarities. Surprisingly, *geomorphology* and *water use efficiency* possess highest overall interdisciplinarity, indicating that these are generally important topics with impact across most drought research topics (Figure 5 and Figure 2). For example, research where
geomorphology and water use efficiency well overlap focuss on soil processes, soil formation and impacts on plant water uptake as well as irrigation. In contrast, *plant genetics* and *crop breeding*, jointly with *plant stress response* are less interdisciplinary with some similarities amongst each other. In this context, *forecasting* is only marginally linked to these three topics (Figure 5 and Figure 2).

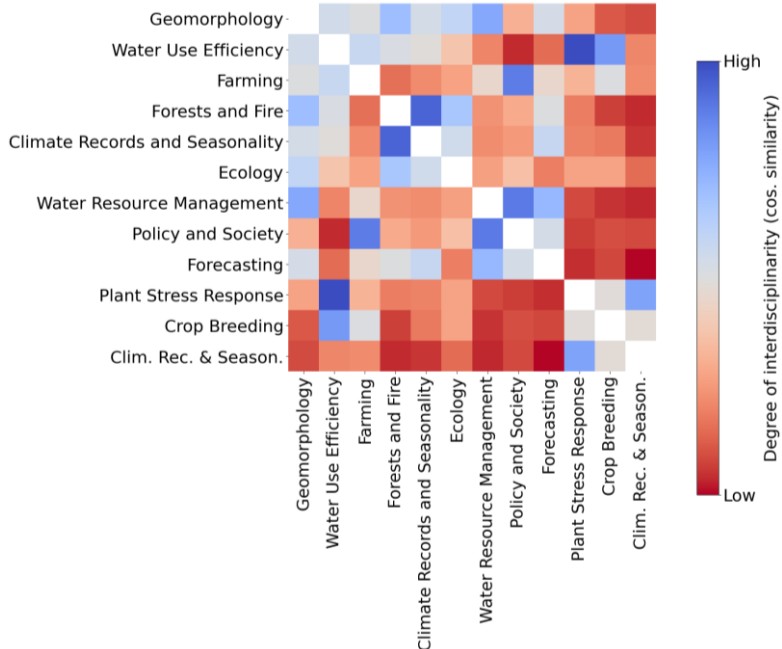

**Figure 5: Thematic overlap of twelve research topics. Cosine similarity shows strength of overlaps between topics. It is the topic-wise similarity score that is the numerical value for cross-topic intersection. Stronger thematic overlaps (e.g., plant genetics and plant stress response) are identified by higher similarity, while minimal similarity score also stands for minimal intersections between two topics (e.g., plant genetics and forests and fire). Topics are sorted from highest overall similarity score (geomorphology) to lowest overall similarity score (plant genetics).**


More specific topics of tier 3 reveal a different picture regarding inter-disciplinarity (Figure 6). These topics are more specific than the tier 2 topics. Here, *climate change*, *knowledge review*, *remote sensing* and *soil processes* lead the field in terms of overall interdisciplinarity (Figure 6). We note that these topics are highly inter-disciplinary based on the topical analysis results. *Knowledge reviews* seems to not only review a specific topic but make inter-connections beyond single topics. *Climate change* does relate to several topics e.g. through cause and impact. *Soil processes* affect a large number of fields apart from the mere research topic of its own. The same for *remote sensing* albeit here, with emphasis on two specific topics: *Meteorological* and *Palmer drought index* which themselves also possess a high similarity index (Figure 6). In contrast to interdisciplinary topics, *flash droughts*, *oil palm*, and *plant species and regional studies* are rather narrow in scope with low overall interdisciplinarity within drought research (Figure 6). Cosine similarity highlights further topics with strong similarity while for other topics we identify major difference and little overlap in terms of content. Here, interdisciplinarity appears to be more challenging rather than an opportunity to form larger content clusters.

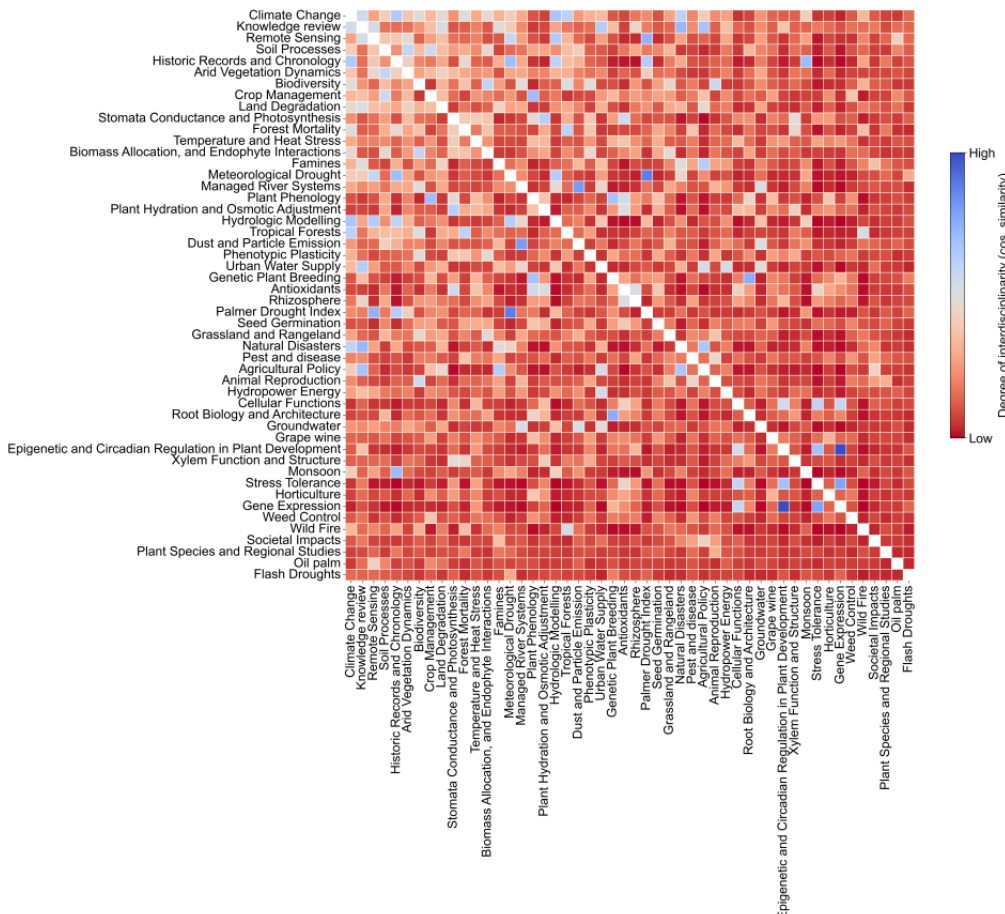

**Figure 6: Thematic overlap of the fifty highly specialized research topics. Overlap of the 50-topic clustering results is based on the cosine similarity between topics as metric for similarity and inter-disciplinarity (high to low). The vertical**

**axis is sorted for highest to lowest overall similarity score. Highest overall similarity score was calculated for climate change and knowledge review. Some topics such as gene expression exhibit very low overall similarity score marked by mostly dark red although there can be at times a strong relation to individual topics (dark blue) for instance gene expression and epigenetic and circadian regulation in plant development.**


The multi-system impact of drought as natural hazard challenges the often topic specific approach of research projects by requiring a multi-system response. The trend-shift in 2007 towards drought research becoming more systemic again may acknowledge this multi-system property of drought hazard, exposure and vulnerability. Until 2007, drought research evolved to become more disciplinary as explained by the annual similarity index (Figure 1). Notable trend shifts occurred in the 1980s

when genetics was introduced into drought research and started to become a major topic in the 30 years following. This led to a first trend shift from becoming more disciplinary by 1983 to again becoming more interdisciplinary with high volatility throughout. The second trend shift happened in the years around 2007, when similarity was lowest and succeeded again from a stable upward trend. Noting the systemic impact of drought, we welcome the trend of drought research to become more interdisciplinary because only systemic approaches can properly enhance drought impact resilience across systems.


### 3.4    Geographic patterns and priorities

Research priorities vary with regard to geographic context. We analyzed continent specific topical signatures in drought research (Figure 7). The largest number of studies refers to Asian (18.0%) and African countries (11.2%) although research budgets in Europe and North America are generally higher than in Africa. This indicates that drought is well recognized as

challenge to many African countries, even more so in Asia. As major pattern, forecasting dominates drought research in geospatial context in all regions. In Africa and Oceania, *forecasting* is closely followed by research on *policy and society*. This pattern is less pronounced for Europe, Asia and South America (Figure 7). In Africa, *farming* is the third largest topic with still 12.5 percent and the other topics are less relevant. North American drought research prioritizes *water resource management*, *ecology* and *forests and fire*, just after *forecasting* with less weight on *policy and society* as compared to the

other regions. In Oceania and Antarctica, *ecology* is the third major topic. Although there are distinct regional differences amongst the geographic regions, the Southern and Northern hemisphere do not show distinct topical patterns (Figure 7). *Forecasting*, and *policy and society* are the two major topics with geospatial context. *Plant genetics* is the major topic for research with no geospatial context, just before *plant stress response* and *water use efficiency*, due to the focus on biological, physiological, genetic and molecular scales.


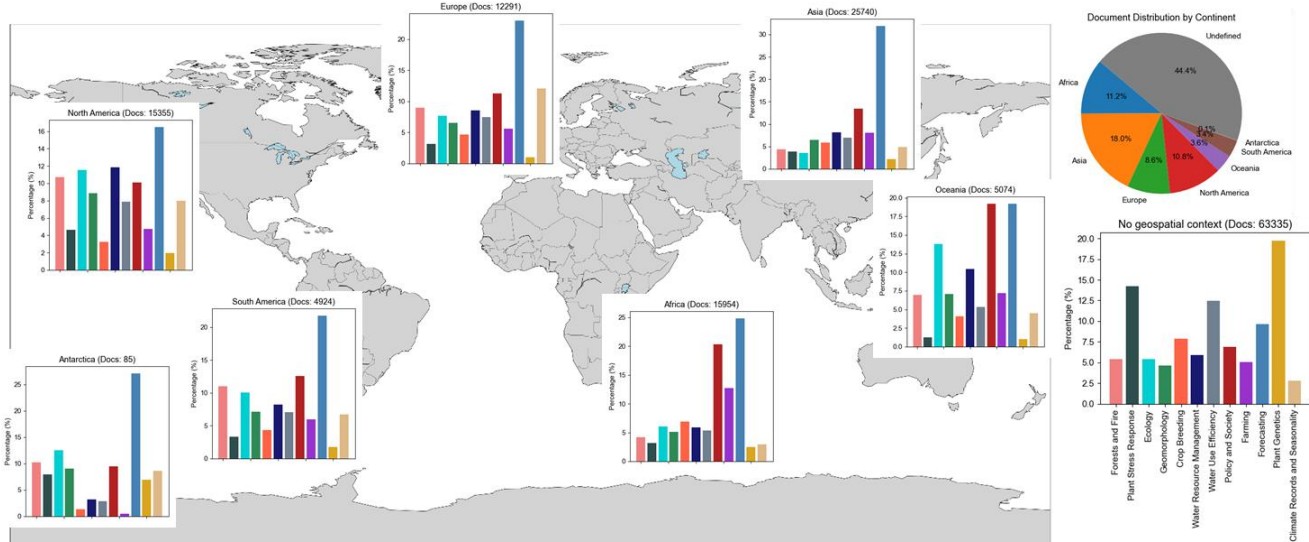

**Figure 7: Geospatial distribution of drought research foci for all years. Distribution of the twelve research topics differs depending on region of the world. Distributions were calculated based on mentioning the name of continents or states. Oceania and Africa exhibit a remarkably high share of policy and society related research. Forecasting is everywhere the topic with strongest weight, in particular in Europe and Asia. In North America forests and fire, ecology and geomorphology are almost at level of forecasting. Plant genetics dominates research of no geospatial reference. For high resolution Figure, please see the Supplement Figure S4.**

## 4 Discussion

This study analyzed over 130,000 peer-reviewed articles on drought research published between 1901 and 2022, identifying key trends, emerging topics, and interdisciplinary shifts within the field. The analysis revealed that drought research has undergone significant transformations, with increasing attention to plant genetics and forecasting methods, while traditional areas like ecology, water resource management, and forestry have seen a relative decline. Regional analysis showed that forecasting methods are a dominant focus globally, whereas policy and societal dimensions play a particularly important role in Africa and Oceania. Furthermore, interdisciplinarity in drought research decreased until 1983, followed by a rise from 2007 onwards, signaling a growing trend toward more integrative approaches. These findings provide a foundation for evaluating the evolving priorities in drought research and their implications for addressing drought risk in diverse contexts.

### 4.1 Definition and use of drought in literature

An important aspect of analyzing the results is understanding the definition, mention, and meaning of the word drought. The definition of drought has been widely discussed in literature. The discussions cover quantitative aspects, such as different drought indicators (Satoh et al., 2021), specific drought events such as flash droughts (Schwartz et al., 2023), and a more

generalized concept of drought as driven by and as threat to societal systems (Mishra and Singh, 2010; van Loon et al., 2016b). In line with these discussions, our results identify flash droughts, the Meteorological Drought Index, and the Palmer Drought Index as distinct, highly specialized topics within the broader drought research literature (see e.g. highly specialized topics in Figure 2). At a more general level, drought is perceived as systemic, encompassing ecosystem, societal, and economic dimensions. At this level, case studies are particularly useful for quantifying connection strengths and impacts within specific environments (van Loon et al., 2019; van Loon et al., 2024). In genetic and plant physiological research, drought is defined as a system state that hampers plant growth (Gaudin et al., 2013; Moran et al., 2017). In plant genetics, highly specialized topics focus on methods to foster drought tolerance and drought resistance of plants through e.g. enhancing water use efficiency.

In agreement with literature, this analysis' results and the topological maps of drought research indicate, that there is not a unique understanding of drought. Rather, the different understandings, scale and concepts are interconnected at both general and specific levels. The results build on the strength of LDA as a method, to calculate connection strengths based on co-occurrence and frequency of multiple keywords and topics rather than focussing on specific terms and meanings. Hence, this method results in maps of connection strengths between different systems, foci and perspectives (Figure 5 and Figure 6).

## 4.2    Forecasting Methods

Forecasting has emerged as a major and increasingly relevant topic in drought research. This growing importance is driven by several factors, including the substantial data requirements, the need for data integration platforms, environmental monitoring systems, and the application of artificial intelligence to generate indices. The identified emerging trends in the highly specialized Tier 3 topics and these factors contribute significantly to the rising prominence of drought *forecasting* (Wardlow et al., 2017; Pasteris et al., 2005). Additionally, recent occurences of rapidly developing and large-scale drought events, such as in the La Plata Basin in South America and West Africa (Geirinhas et al., 2023). Highlight the need for enhanced mechanistic understanding and forecasting capabilities. Improved forecasting will improve the readiness to manage more frequent drought conditions and support food security for a growing global population (Krishnamurthy R et al., 2022).

Despite the increasing application of machine learning approaches in drought forecasting (Al Mamun et al., 2024; Prodhan et al., 2022), our analysis did not identify machine learning and artificial intelligence as distinct topics. Similarly, early warning systems (Funk et al., 2019) and compound events (Ridder et al., 2022; Yin et al., 2023; Lesk et al., 2022) were not identified as distinct topics although urgency for progress in these topics is perceived (FAO-WFP, 2020). There are two possible explanations for this. First, these topics may not have emerged as distinct areas within the drought research domain and could be distributed across broader research themes. Second, the volume of research on these topics may be insufficient to form distinct clusters. To make a meaningful impact on drought research, these emerging topics need to gain further momentum through increased publication numbers, research focus and funding mechanisms.

### 4.3    Bridging the Arch of Drought Research

Given the challenges in food and water security, it is now the time to bridge the gap between forecasting drought impacts e.g. under climate change and the consideration of genetic advances in drought tolerance for agricultural production. However, it 350 is particularly challenging to bridge this gap given the distance in the arch of drought research and current similarity between plant genetics and forecasting (Figure 2). Some projects already address this challenge by aiming to include genetic variability of plants in crop models or use crop growth models for identification of climate adapted varieties (Parent and Tardieu, 2014; Chenu et al., 2017). Hence, crop yield forecasts under climate change scenarios must give stronger consideration to genetic advances and plant molecular processes than currently is explored (Stella et al., 2023). Nevertheless, increasing drought crop's 355 drought resilience and forecasting will not suffice as sole mechanisms. Other mitigation strategies at political and ecosystem level can provide equally important and case specific solutions. Knowledge reviews are one tool and required to bridge interdisciplinarity which they already do as illustrated by reviews leading the overall interdisciplinarity score (Figure 6).

### 4.4    Implications for research, policy and institutions

These findings have significant implications for scientific community, policymakers and institutions addressing drought issues. The current topical research emphasis lies on drought forecasting methods and plant genetics (Figure 4). Both topics guard food security and imply food security as human priority for funding and research interest. Plant genetics provides methods to identify genes and produce variants with higher drought tolerance by altering a variety of physiologic processes (Figure 2). Forecasting explores methods to forecast drought with regard to risk monitoring often related to agricultural impacts and 365 drought indicators and for specific events (Figure 2). In contrast, topics that address drought impact on socio-ecosystem compartments such as *ecology* and *water resource management* held greater importance in the past compared to their current relevance. Policy and funding agencies must decide based on their priorities whether this is a desirable trend or not, and align funding strategies accordingly.

### 370    4.5    Limitation of Latent Dirichlet Allocation

The data for this research was limited to abstracts, titles, and keywords. While this approach allowed for a large volume of articles to be analyzed over an extended temporal period, it comes with certain limitations. The primary advantage is that abstracts, titles, and keywords are well standardized and harmonized, enabling the inclusion of a vast number of articles. However, a key disadvantage is the uncertainty regarding how accurately these abstracts and keywords reflect the full content 375 and findings of the articles. The results are heavily dependent on the effectiveness of the peer-review process, underscoring the critical role of abstracts in research communication.

A more in-depth study, potentially utilizing more advanced models or normalization procedures, would require additional computational resources (e.g. compare (Callaghan et al., 2020; Ogunleye et al., 2023). For instance, normalizing word

frequency could help account for differences in document length. It's important to note that a full-text analysis might reveal that specialized research is more interdisciplinary than it appears from abstract-only analysis. Additionally, the use of semantic analysis could extract further insights and generate more detailed information (Geeganage et al., 2024; Niu et al., 2022).

## 5    Conclusions and Future Directions

This study offers a comprehensive data-driven topology of drought research, providing valuable insights for decision-makers, researchers, and institutions. By mapping the current topical priorities, geographical distribution, emerging trends, and interconnections between research areas, we contribute to the development of systemic drought resilience frameworks. These frameworks ideally embrace the increasing interdisciplinarity observed since 2007, ensuring a holistic approach to drought resilience (Hagenlocher et al., 2023). Key topics to consider in this context include *climate change impacts*, *reviews,* and

*remote sensing* which exhibited highest overall interdisciplinarity – and are potentially critical to the effectiveness of systemic drought resilience frameworks (Hagenlocher et al., 2019). Additionally, central components of drought research, such as water resource management – essential for societal applications like drinking water supply, cooling, and irrigation—must be integral to these frameworks (Jasechko et al., 2024).

While this analysis highlights current trends, it also identifies areas that are potentially overlooked in drought research. Future research directions should also focus on underrepresented areas, such as the integration of drought impacts on less-studied ecosystems, which are critical to the Earth's life support system (IPCC, 2021). Furthermore, the socio-political implications of water scarcity and the development of more localized and culturally sensitive drought resilience strategies deserve greater attention. Notably, the study did not identify machine learning approaches, studies of compound events, and early warning

systems as distinct topics, despite their recognized importance in the field. Addressing these gaps will ensure that future drought research not only follows trends but also explores crucial areas that are currently underrepresented, leading to more comprehensive and effective strategies for mitigating drought impacts.

The trends, geographic patterns, and connection strengths between topics revealed in this study serve as a tool to guide the

development of future drought resilience frameworks. While much of the drought research falls into highly specialized areas, our results highlight where topical connections between these topics are strong and where they are weaker.

Although this study is grounded in natural language processing of a large scientific corpus, it does not replace a qualitative assessments of scale and event-specific impacts. Instead, it offers a broad overview that complements more specialized

research by revealing the overarching connections and trends within drought research. Decision makers, policy institutes and researchers working on response strategies to address drought issues potentially do well by considering the full breadth of outlined topics.

**Code availability**

Supporting information and code availability is available upon reasonable request from the corresponding author.

**Data availability**

The abstracts' copyright remains with Elsevier B. V. (Elsevier). Permission for research usage was granted exclusively to the Leibniz Centre for Agricultural Landscape Research (ZALF) e.V., prohibiting any redistribution of the underlying text data. Sharing is limited to text and data mining results upon reasonable request, adhering to the © Some rights reserved stipulation. Usage is approved solely for academic research, distribution, and reproduction given the original author and source are duly acknowledged.

**Author contribution**

Conceptualization, formal analysis and writing – original draft preparation::RB, EER. Writing – review and editing: RB, GG, MH, CN, AT, and EER. All authors have read and agreed to the submitted version of the paper.

**Competing interests**

The authors declare that they have no conflict of interest.

**References**

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

## 6    Appendix

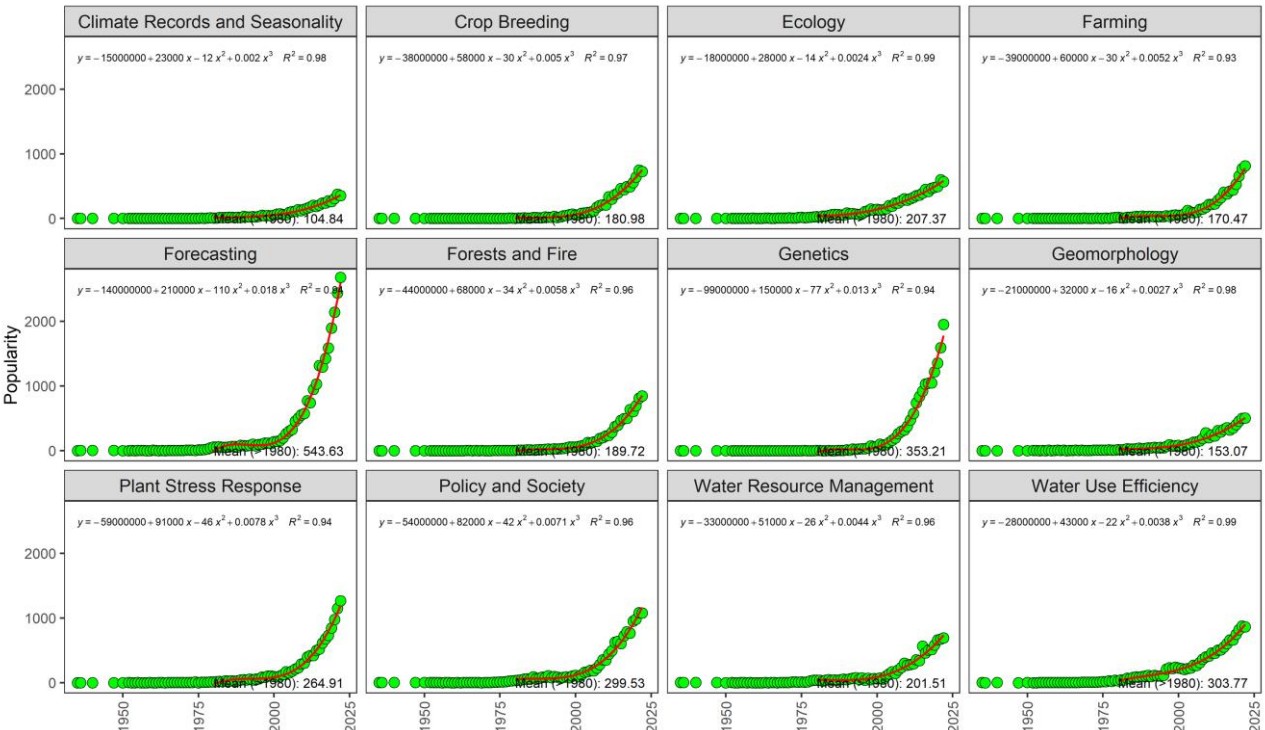

**Figure A: Data and trends of total annual publications. Data and polynomial trends of annual publications by major topics per document. Popularity is the number of publication on the particular topic with the polynomial fitted function to indicate the trend since 1980 to 2022 and the mean for this period is given.**

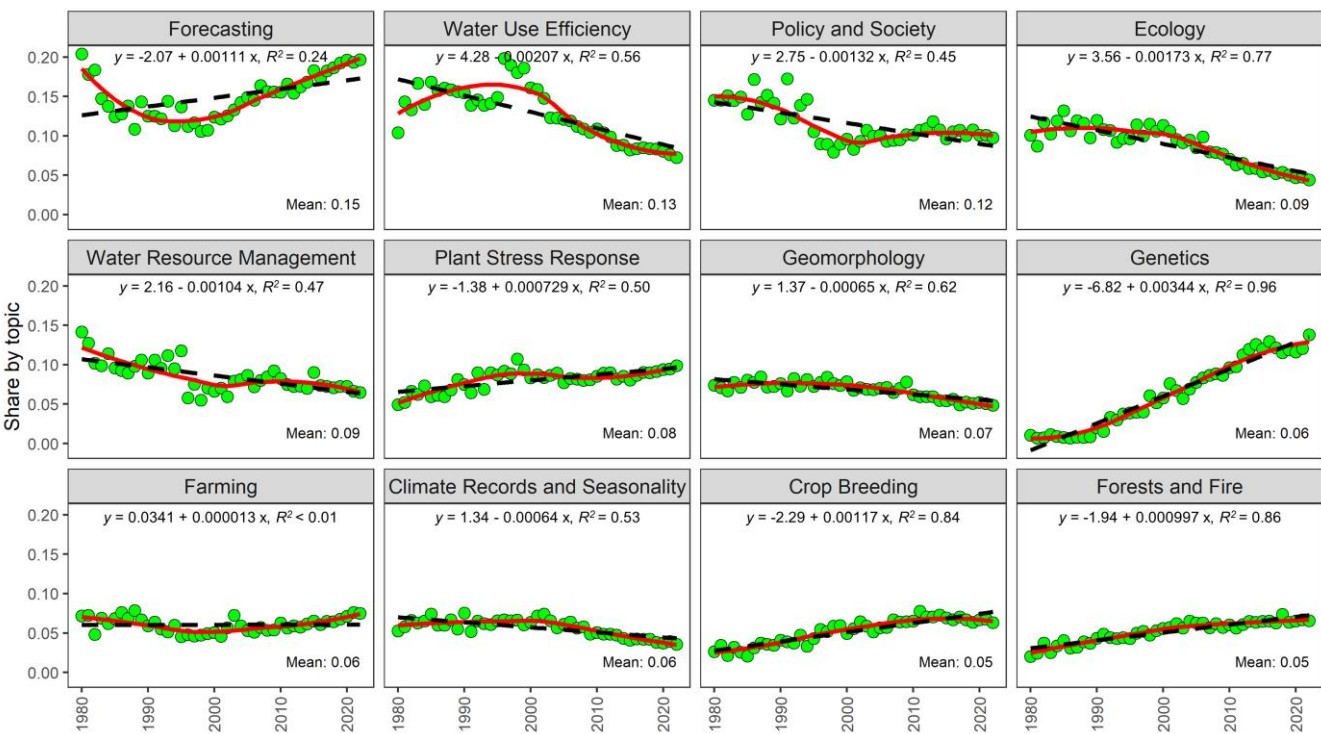

**Figure B: Annual share of topics to drought research. Annual shares of research topics ranked by mean annual share. The formula for linear regression provides the trend over the last four decades (black dashed line). Genetics has the highest positive trend with highest Pearson correlation coefficient. Total publication growth is exponential while annual share refers to the single year, which leads to forecasting and genetics as the topics with highest overall share.**