# Peer review of "Drought Research Priorities, Trends and Geographic Patterns"

_EGUsphere, 2024_

## Author Comment (AC1)

Black text: Reviewer #1.

Red text: Response by the authors.

**1. Summary**

This research article analyzes over 130,000 peer-reviewed articles on drought research from 1901 to 2022. It highlights a shift in research priorities towards plant genetic research for drought-tolerant genotypes and methods in drought forecasting, with decreasing focus on ecology, groundwater, and forest research. The study underscores the importance of interdisciplinary research and recommends enhanced interdisciplinary and systemic approaches to address drought as a multi-sectoral risk.

Thank you for the positive overall evaluation of the manuscript. We appreciate your time and effort in reviewing the manuscript. In the forthcoming revision we will consider each of your suggestions and implement the necessary changes.

**2. General comment**

This work undertakes an impressive effort by conducting a review of drought-related literature spanning over a century. The emphasis on interdisciplinary approaches to address multi-sectoral drought risk deserves particular attention. I found the section discussing the geographical distributions of the topics extremely interesting. While the article is well-written, improvements could be made by addressing certain aspects outlined below, such as, but not limited to, providing more clarifications and references throughout the manuscript, improving explanations in the figure captions, adding additional references to support statements, and ensuring consistency in numerical representation.

We will take up these suggestions in the review process.

**3. Specific comment**

**1 Introduction**

Lines 36-37: Might be relevant to additionally reference here the recent preprint looking at drought as a continuum process: https://egusphere.copernicus.org/preprints/2024/egusphere-2024-421/

We will consider this reference where appropriate.

Lines 40-50: the line 38 provides examples of drought-sensitive sectors, followed by a detailed discussion of impacts on ecosystems and agriculture in the next paragraph (lines 40-50). However, it does not delve into the impacts on health, energy, and socio-political stability mentioned earlier as well.

We will add information on these impacts, as they are also touched on later.

Lines 51-57: this paragraph concludes the introduction by emphasizing the importance of adaptation strategies and the value of long-term drought research. I think it could benefit from better highlighting this paper's contribution. I think it could benefit from better highlighting this paper's contribution. For example: "The long-term research undertaken by this work helps reveal patterns and gaps in our understanding of drought, enabling the development of more effective adaptation strategies." This would underscore the unique insights and contributions of the paper to the field.

**2 Methods**

Line 60: are they all scientific publications (the 131,748 abstracts)?

Yes, as stated by the source.

Lines 73-74: while describing the "three Tiers" here for the first time, it would be helpful to refer to Fig. 2 already for a clearer schematic representation. This visual aid will enhance understanding and provide an easier reference for readers.

We will add reference to Fig. 2

Line 77: I would emphasize already here in the beginning that the similarity between 2 topics is **within a document** otherwise its not clear until the reader reads about it on Line 79.

Thank you. We will clarify this earlier as suggested.

**3 Results**

**3.1 Major and specific topics in drought research**

Figure 1: I think this Figure needs a clearer explanation in the caption, stating what exactly is shown on the right x axis and the left x axis (especially on the right – the similarity index corresponds to the degree of intedisciplinarity).

We will clarify the axis.

Figure 2: the figure would benefit from adding labels such as "major topics" next to Tier 1 with an indication of "5 topics," "intermediate topics" next to Tier 2 with "12 topics," and "highly specialized topics" next to Tier 3 with "50 topics." This would help readers better orient themselves within the manuscript.

Thank you. We will take up this suggestion.

Figure 2: the categorization of topics (M,M,I,M,E) lacks clarity, particularly the selection of "E" (events and historical analysis) for the topic "Precipitation and Drought types." Why wouldn't "Methods and processes" be a fitting category for this topic?

We will reconsider and clarify this.

Lines 117 and 119: require consistency in numerical representation, whether as spelled-out words ("twelve topic") or numerals ("12-topic").

Well spotted. We will clarify this here and throughout the manuscript.

Figure 3: a question regarding the sequence of the 12 topics displayed in the box: should they align with the sequence seen in Tier 2 of Figure 2? Additionally, it would enhance clarity to label the box containing the 12 topics as "Intermediate Topics Tier 2" at the top, facilitating a quicker understanding of its connection to the preceding Figure 2.

They should not align with the sequence in Tier 2. As in Figure 2, we will add a heading to the legend to facilitate the interpretation of the Figure.

Lines 144-146: quite hard to read this sentence, I would recommend rephrasing it.

We will revise the sentence accordingly.

**3.2 General and emerging trends**

Line 168: would recommend to add ref to Figure 4 here already? And potentially to explain why to focus only on last 4 decades?

Agreed. We will add the reference here already.

Line 178-179: the authors should provide references to support this statement. Additionally, it's crucial to note that while increased drought resilience of crops through genetics is valuable, relying solely on this aspect may not suffice, and other mitigation strategies are equally important.

We will take this point up in the manuscript.

Figure 4: The color choice for "Plant genetics" seems to have changed and become more orange as compared to more yellow assigned to it in Figure 2.

Indeed. We will modify Figure 4 to match the color code throughout the manuscript.

Lines 191-194: one of the contributing factors is also the technological advancements that have strengthened forecasting methods.

Thank you. We will note this in the manuscript and add references accordingly.

**3.3 Interdisciplinarity of drought research**

Lines 208-209: "manifested in sedimentary records and tree ring records" is it referring to "Forest and Fire topic? Would recommend to mention it in the text.

We will clarify this sentence.

Line 236: what exactly does "regional studies" refer to? Isn't it more about spatial characteristics rather than a specific topic?

We will investigate this in more detail and clarify the result in the manuscript.

Figure 6: I'm surprised to find "Drought Palmer Index" listed as a distinct topic rather than under a broader category like "Drought Index." Additionally, I couldn't find anything related to Forecasting among these 50 topics. Could it be that certain topics, such as Forecasting, have been split into smaller segments (e.g. modelling)?

Yes, this is the case. We will clarify the focus on smaller subtopics for Figure 6. The meteorological drought index and Palmer Drought Index are both amongst the 50 sub-topics. The major general topic forecasting has been split into smaller topics. Forecasting as the more general topic is made up of many smaller topics which are also inter-linked. Modelling and the two drought indices are three out of the fifty more focused sub-topics.

**3.4 Geographic patterns and priorities**

Figure 7: Is it built on 40 or 100 years review, need to mention this. Additionally, wasn't "forecasting" previously indicated in blue but now appears in yellow? Consistency in color usage throughout the manuscript is recommended. Regarding the figure caption, isn't "forecasting" positioned at the top everywhere, not only in Europe and Asia (same as mentioned in the abstract).

Fully agreed. We will harmonize the color code for the 12 topics. We will clarify that forecasting is positioned at the top i.e. of major interest everywhere.

**4. Discussion and Future Directions**

**4.1 Implications for research, policy and institutions**

Line 288: its not clear which indicators are meant here? Drought indicators or agricultural impact indicators?

We will clarify. Drought indicators are meant.

Lines 288-290: what about the priorities of research community as well and their influence on the trend?

An individual's H-Index of 50 can be considered high while on average the 50 sub-topics possess 2620 publications. Looking at the number of publications (>131k) and the potential size of research communities, individual research communities must be of considerable size in order to play a role in the formation of topics.

Line 291: to advance drought impact forecasting, it's crucial to consider several related topics besides genetics.

Agree. This paragraph and discussion should focus on bridging the gap. While many related topics are considered already, plant traits, genetic breeds and advances seem to be considered rather limited.

Line 298: "knowledge reviews **as** are required …" spelling

This will be corrected.

Line 303: here written as "drought indices" but its only "Palmer Drought Index?" within the 50 topics?

We will clarify that also the meteorological drought index is a separate topic within the 50 topics.

**Supplementary**

Couldn't find Figure A and Figure B (Lines 187 and 190) in the Supplementary, I only see Figures S1, S2, S3, S4

This is correct. We will correct the cross-references in Section 2 to the Appendix instead of Supplementary.

---

## Author Comment (AC2)

Black text: Reviewer #2.

Red text: Response by the authors.

The article describes an analysis of bibliographic information (abstracts, keywords, title, publication year, language) performed for the period 1901-2022 to identify trends and focus areas in drought research. In general, the article is well written, contains high quality figures for illustration and addresses an interesting topic. I find in particular the consideration of trends and regional priorities in drought research very interesting. However, I have a number of concerns and suggestions that may be considered to improve the manuscript:

Thank you for the positive overall evaluation of the manuscript. We appreciate your time and effort in reviewing the manuscript. In the forthcoming revision we will consider each of your suggestions.

**General comments:**

The methods applied by the authors are based on a number of important assumptions that should be made transparent to the readers and also being discussed. First of all, the authors limit their analysis to abstract and keywords. This has the advantage that a large number of articles can be considered but the disadvantage, that it is not clear at all how well the specific abstracts and keywords really reflect the content of the articles and the research results.

We will address and state this limitation in the discussion. The limitation of only looking at the abstracts and not at the full paper is well noted. While a dependency, the methods and results rely on and benefit from the peer review process carried out on abstracts.

Second, it assumes, that terms and definitions used in drought research have been similar across time, regions, disciplines and scales.

This is correct to some degree. We will address this point in the discussion section. The strength of Latent Dirichlet Allocation (LDA) is that it uses all words within a document for analysis. While we cannot exclude terminology to play a role in topic formation, it is always a number of keywords rather than few which form the context and a single topic. We will add this in the discussion.

Certainly, these are strong assumptions. To my experience it is frequently not clear whether statements in the abstract section represent the present understanding in the research community, opinions of the authors, or really conclusions drawn from the research results that are justified by real data. One example is the abstract of the present manuscript. In the last sentence the authors state: "In future, we recommend research and funding agencies to strengthen the track of more interdisciplinary and systemic cross-topic drought research in order to cope with drought as a multi-sectoral risk requiring multi-sectoral response frameworks." Is this recommendation justified by the results of the present analysis? Certainly not. Instead it is the opinion of the authors and reflects well a popular believing of a large part of the drought research community. Nevertheless, this statement can also be challenged. Interdisciplinary, systemic, cross-topic and multi-sectoral research is more expensive and requires more effort than targeted disciplinary research. Funding agencies may have to decide therefore in practice whether they fund 5 small disciplinary projects or one big interdisciplinary project. It seems therefore, that such a general statement is not appropriate.

Certainly, to provide solutions for very specific problems (that nevertheless can have a big impact) such as to improve drought tolerance of crops, to develop more efficient water use technologies or to develop new approaches to improve groundwater recharge, disciplinary research will likely be sufficient. In contrast, to improve drought management in large basins, a multidisciplinary approach

may be appropriate. This example shows that the approach to simply extract words and terms from the abstract without considering the other sections and the context of the article may result in misleading conclusions.

First, we agree with reviewer #2 that some abstracts in the corpus may not reflect the content of the paper or may also include misleading statements. This is a qualitative aspect of the source data which sets constrains to the use of literature and is beyond the scope of our analysis.

Second, we will revise the statement(s) in this study's abstract. The results of the study allows to identify research topics where drought research is already spanning disciplines and sectors, and which sectors are rather looked at disciplinary. We will highlight that this study gives a comprehensive overview across the various sectors affected by drought and where research priorities are based on published articles. Last but not least, we will highlight the importance of addressing water scarcity i.e. drought research through consideration and active inclusion of multiple stakeholders i.e. interest groups in competition for the limited resources. Here, the study quantifies research and sectors affected in Figure 3.

One interesting result of the present study is that the majority of drought related articles address either plant genetics and physiology or drought forecasting. These domains address completely different scales and, even more relevant, may use different definitions what a drought is. At large spatial scales, the terminology is distinguishing drought as an extreme event from aridity as a climate indicator. At plant level drought stress is usually when the water supply is lower than needed for optimal growth. Here, no difference is made between water deficits caused by the fact that crops grow in arid regions and water deficits caused by droughts. Consequently, drought has a different meaning and drought research has another context at these two scales. 3 out of the 12 topics presented in figure 3 (plant stress response, crop breeding, plant genetics) are in a context where drought is understood as a water deficit in general, 7 topics are in contexts that consider drought to be an extreme event and 2 topics (farming, water use efficiency) may use one or the other understanding, depending on the context. While it is as it is, this example shows that the use and meaning of terms may be context specific, a difference that likely cannot be detected by a simple analysis of article abstracts.

I also think that the use of terms in the literature has changed over time. For example, most of the large body of historical articles related to dryland research will have drought in the list of keywords although, according to the present understanding, this research is more linked to aridity and therefore not to drought as an extreme event. Consequently, the results of the trend analysis performed by the authors also have to be taken with care.

Thank you for pointing this out. This is an important finding. We appreciate reviewer #2's analysis and will expand on the definition and meaning of drought in different contexts. We will investigate and note in the discussion, whether the observation that drought can be defined as water deficit in general (plant physiology related) and as an extreme event (forecasting related), and if these two definitions are overlapping. This related to the notion that the use of drought may have changed over time. We will also note this in the discussion.

**Specific comments:**

**Abstract:** see general comment 1

**Introduction:** is very general and mainly describes why drought research itself is relevant. Research questions are not mentioned and the reader will not be prepared for the analysis described later. There is hardly any link between the introduction and the methods section. Consequently, the reader does not know what the authors what to find out and why they use the specific methods described in

the Methods-section. The specific gap of knowledge is not described so that it is difficult for the readers to see the novelty. There are so many review papers in the literature about drought research, please make more clear what the innovation of the present article is and what you make differently compared to the review papers that have been published before.

Thank you. We agree and will strengthen the link between introduction and methods. We will achieve this through a paragraph on motivation, addressing the points made by Reviewer #2.

**Methods:** It is well described which methods have been used but not why. What is the advantage of using LDA in that specific case, compared to other alternatives?

We will address this question in the method section.

**Results:** The first four lines are a summary of the methods but not results.

Thank you. We will remove the summary of the methods from the results section.

Figure 1: Many of the readers will not know what cosine similarity means. Please explain that in the figure caption. That the number of drought related articles has increased over time is not surprising since there are more researchers who write more articles per year, compared to former times. I would find it more interesting to see the change in the relative share in drought related articles in the whole scientific literature.

This is a good point and we will add information on drought research in comparison to the overall number of articles published for example in the Web of Science Core Collection. We will also add the explanation for cosine similarity to the figure caption.

**Discussion:** Please add one section describing the limitations of data and methods used in the present research.

Details on limitations of data and methods will be added either as discussion section or in the methods section.

---

## Author Comment (AC3)

Black text: Reviewer #3.

Red text: Response by the authors.

The authors use a well-known method to give an overview of drought research topics and their development over time. While the method is not new, I find the results and the added thoughts by the authors to be of interest to the research community.

In order to make a topic model research overview interesting and relevant the analysis needs to go beyond reporting and interpret the conclusions for the research community. The authors do this very well. For example, I really like the discussion on including plant genetics into future yield prediction in Section 3.2.  My only criticism would be where these interesting discussions currently happen. At the moment these very relevant findings and interpretations are hiding at the end of the individual results sections and are only briefly picked up in the discussion section. I would recommend a clearer split between results and discussion, meaning that e.g. Lines 191 to 196 should be part of the discussion, not the results. Same with other paragraphs. E.g. the discussion around missing topics in L156 to 164 then repeats in L301 to 306.

Thank you for your positive evaluation of the manuscript. We appreciate your time and effort in reviewing the manuscript.

We agree that the paper will benefit from restructuring. We will henceforth move discussion points from the results section to the discussion section, which is where they belong to.

The discussion section on drought resilience frameworks makes an important point (L308). But at the moment it feels more like an afterthought. It could, for example, already be introduced in the introduction as a motivation for a drought research overview and then elaborated more in the discussion.

Thank you. We will stronger integrate drought resilience frameworks already in the introduction and motivation.

Minor points:

L18: Please already mention that the generative model you are using is LDA, e.g. "using a generative model (topic model)", as for some people that term will be more familiar.

As suggested, we will mention LDA already here.

Introduction: Please expand on the motivation for this study. The first three paragraphs repeat well-known information that summarises why drought is a relevant topic. This could be shortened into one paragraph. And then the introduction should introduce the motivation for this specific analysis. Why do we need an analysis of drought research topics? (relevant for drought resilience frameworks?). What is the problem with past drought reviews? (Too many publications to humanly read? (Stein et al, 2022)). There are a lot of bad scientometric reviews out there. What makes this one different?

We will expand on the motivation for this analysis thereby address the questions pointed out.

L64: Scopus is not the largest database (see https://help.openalex.org/coverage for a comparison).

Thank you. We will name other databases for scientific literature and remove that Scopus is the largest one.

L70: "LDA did cluster" is a strange grammar. Consider rephrasing.

We will rephrase the sentence.

L96: The description of the methods repeats here. Consider shortening.

We will remove the repetition of methods from results section.

L129: Can you add a range? How many papers did the smallest/largest topic include?

We will include ranges in the text.

L133/140: This might be an editing problem, but there should not be a paragraph break between these two, as the thought continues. In the next line.

We will move the positioning of the Figure to fit the flow of content.

L152: It would be good to see which topics the author classifies as impact, event and methods related. I would this categorisation also helpful in structuring Figure 6. However, I would understand if a clear categorisation for some topics is difficult.

We will more prominently point to the classification detailed in Figure 3. We will also consider to add the classification in Figure 6 axis labels.

L161: A keyword search across the articles might give an indication which of the two explanations are true.

We will consider the keyword search in the revision and report the findings in case they are meaningful.

Section 3.2. I think the labels for Figures A and B have been switched for the entire section.

Well observed. We will switch the cross-references to Figures A and B accordingly.

Figure 5/6. The upper triangle of the plot only mirrors the lower triangle. Consider removing.

This is correct. We considered removing. However, it makes the figure more accessible, if the labels are sorted by overall similarity and scores can be read line by line, with full cross-reference to the second axis labels. Hence, we suggest to keep both triangles.

L213: Can you elaborate on the surprising geomorphology and water use efficiency connection? In what kind of research do they overlap?

We will identify and explain the overlap for both topics.

L235: This is the only discussion I did not really like. It sounds a little bit like you are advocating for interdisciplinarity for interdisciplinarity' sake, which I do not agree with. Plus you are only looking at

abstracts. A full-text analysis will likely reveal the specialised research to be more interdisciplinary as well.

We will note that a full-text analysis will likely reveal stronger overlaps.

We will also better explain why interdisciplinarity is crucial for drought research. Here we will stronger draw on the results of the study, the widths of topics in research affected by water scarcity, and the competition and limitation of water resources availability – in particular during drought periods.

Section 3.4. The low numbers of drought research in North America are surprising and need to be checked. The US is always a major contributor in previous research distribution analyses (e.g. Emmer (2018) and Callaghan et al, 2021). Particularly for drought research, I would expect a lot of research with the megadrought in California in recent years.

This is surprising, indeed. One reason is the volume of research funding in North America. In contrast to Emmer 2018, we did not attribute geographic location based on (co-)authorship affiliation. We have chosen a similar approach as Callaghan et al. (2021) by using geographic information i.e. country names and for USA federal state names to assign geographic context. We will check again the consistency, point out this surprising finding in context to additional references and findings, and elaborate potential reasons in the discussion.

L306: Is it possible that the diversity of terms in use for these topics might prevent a clear topic (e.g. compound vs multi-hazard vs hazard cascade…)?

The strength of LDA is that it uses all words within a document for analysis. While we cannot exclude terminology to play a role in topic formation, it is always a number of keywords rather than few which form the context and a single topic. We will add this in the discussion.

 Callaghan, M., Schleussner, C. F., Nath, S., Lejeune, Q., Knutson, T. R., Reichstein, M., ... & Minx, J. C. (2021). Machine-learning-based evidence and attribution mapping of 100,000 climate impact studies. Nature climate change, 11(11), 966-972.

Emmer, A. (2018). Geographies and scientometrics of research on natural hazards. Geosciences, 8(10), 382.

Stein, L., Mukkavilli, S. K., & Wagener, T. (2022). Lifelines for a drowning science-improving findability and synthesis of hydrologic publications. Hydrological Processes, 36(11), e14742.

---

## Author Response (AR1)

**Reply to Editor-in-Chief**

Black text: Editor

Red text: Response by the authors.

**Editor Decision**

I'd like to thank all three reviewers for the constructive and robust reviews, as well as the authors for their positive attitude towards critical comments that are only to serve the improvement of their very own manuscript.

I agree that this is a cool research using text mining (corpus) which has been widely used in computational linguistics for decades and is a well-known and -established research direction and tool. I also agree that this research has important implications for trend discovery. But more importantly for trend setting for future directions of hydrology and earth sciences - I made a suggestion in the last paragraph on this.

I'd like to ask the authors to address all the comments made by the reviewers, including (but not limited to) adding discussion points related to assumptions, limitations, plant vs forecasting, and all the other major comments. Fleshing out these discussion points about the uncertainties of your research, honestly and transparently [citing Keith Beven here], enhances the quality and credibility of your work by nuancing the take home messages you're communicating to your readers.

Thank you for agreeing and leading the review of the manuscript. We appreciate your time and effort in serving as editor and managing the review process.

We addressed the reviewers suggestions and modified the manuscript to take up most of the reviewers comments. Accordingly, the revised submission includes individual response letters to each of the three reviewers.

Please also find responses to your suggestions below:

In the new discussion section, we added five sub-sections to the manuscript, including a section on:

"4.1    Limitation of Latent Dirichlet Allocation",

where we elaborate on limitations, advantages and further possible methods for topic modelling.

"The data for this research was limited to abstracts, titles, and keywords. While this approach allowed for a large volume of articles to be analyzed over an extended temporal period, it comes with certain limitations. The primary advantage is that abstracts, titles, and keywords are well standardized and harmonized, enabling the inclusion of a vast number of articles. However, a key disadvantage is the uncertainty regarding how accurately these abstracts and keywords reflect the full content and findings of the articles. The results are heavily dependent on the effectiveness of the peer-review process, underscoring the critical role of abstracts in research communication.

A more in-depth study, potentially utilizing more advanced models or normalization procedures, would require additional computational resources (e.g. compare (Callaghan et al., 2020; Ogunleye et al., 2023). For instance, normalizing word frequency could help account for differences in document length. It's important to note that a full-text analysis might reveal that specialized research is more interdisciplinary than it appears from abstract-only analysis. Additionally, the use of semantic

analysis could extract further insights and generate more detailed information (Geeganage et al., 2024; Niu et al., 2022)."

In addition to the reviewers point, I'd like to encourage the authors to also reflect on what aspects of drought research do they think are overlooked in the current trends. This can be in a sub-section of future research direction or a few bullet points in the conclusion. This is a strategic insight for the future researchers to know, besides what's trendy nowadays, what research questions or variables or topics are missing or overlooked currently.

We added and highlighted aspects that are overlooked in the discussion sections, and in the section "5 Conclusions and Future Directions

[…] While this analysis highlights current trends, it also identifies areas that are potentially overlooked in drought research. Future research directions should also focus on underrepresented areas, such as the integration of drought impacts on less-studied ecosystems, which are critical to the Earth's life support system (IPCC, 2021). Furthermore, the socio-political implications of water scarcity and the development of more localized and culturally sensitive drought resilience strategies deserve greater attention. Notably, the study did not identify machine learning approaches, studies of compound events, and early warning systems as distinct topics, despite their recognized importance in the field. Addressing these gaps will ensure that future drought research not only follows trends but also explores crucial areas that are currently underrepresented, leading to more comprehensive and effective strategies for mitigating drought impacts.[…]"

**Reply to Reviewer #1**

Black text: Reviewer #1.

Red text: Response by the authors.

**1. Summary**

This research article analyzes over 130,000 peer-reviewed articles on drought research from 1901 to 2022. It highlights a shift in research priorities towards plant genetic research for drought-tolerant genotypes and methods in drought forecasting, with decreasing focus on ecology, groundwater, and forest research. The study underscores the importance of interdisciplinary research and recommends enhanced interdisciplinary and systemic approaches to address drought as a multi-sectoral risk.

Thank you for the positive overall evaluation of the manuscript. We appreciate your time and effort in reviewing the manuscript. In the forthcoming revision we will consider each of your suggestions and implement the necessary changes.

**2. General comment**

This work undertakes an impressive effort by conducting a review of drought-related literature spanning over a century. The emphasis on interdisciplinary approaches to address multi-sectoral drought risk deserves particular attention. I found the section discussing the geographical distributions of the topics extremely interesting. While the article is well-written, improvements could be made by addressing certain aspects outlined below, such as, but not limited to, providing more clarifications and references throughout the manuscript, improving explanations in the figure captions, adding additional references to support statements, and ensuring consistency in numerical representation.

We took up these suggestions in the review process. Below, we elaborate on the changes made.

**3. Specific comment**

**1 Introduction**

Lines 36-37: Might be relevant to additionally reference here the recent preprint looking at drought as a continuum process: https://egusphere.copernicus.org/preprints/2024/egusphere-2024-421/

We will introduced the reference appropriately, not only here:
"This led to the reconsideration of the definition of drought as rather a being a process than a system state (van Loon et al., 2016a; AghaKouchak et al., 2021; van Loon et al., 2024)."

Lines 40-50: the line 38 provides examples of drought-sensitive sectors, followed by a detailed discussion of impacts on ecosystems and agriculture in the next paragraph (lines 40-50). However, it does not delve into the impacts on health, energy, and socio-political stability mentioned earlier as well.

We agree and added:
"Drought also impacts human health (Vins et al., 2015) e.g. through reducing stream flow, increasing concentration of pathogens, enabling some vector-borne diseases (Cann et al., 2013) and as risk factor of child undernutrition in particular in low-income conditions (Belesova et al., 2019). The impacts of drought are scale specific, event specific and often difficult to quantify due to their

indirect and often systemic character - affacting not only human health and agriculture but also energy and social systems (van Loon et al., 2019; Blauhut et al., 2015)."

Lines 51-57: this paragraph concludes the introduction by emphasizing the importance of adaptation strategies and the value of long-term drought research. I think it could benefit from better highlighting this paper's contribution. I think it could benefit from better highlighting this paper's contribution. For example: "The long-term research undertaken by this work helps reveal patterns and gaps in our understanding of drought, enabling the development of more effective adaptation strategies." This would underscore the unique insights and contributions of the paper to the field.

We added a paragraph on motivation:

"This study is motivated by the need to enhance our understanding of the evolving landscape of drought research, particularly in light of the escalating challenges posed by climate change and water scarcity. While previous reviews outlined the need to synthesize the immense body of literature on drought research (Stein et al., 2022), our analysis distinguishes itself through the use of a data-driven, unsupervised machine learning approach to examine over 130,000 peer-reviewed articles. By exploring long-term research trends, we identify critical shifts in thematic focus, fundamental and emerging trends, and interdisciplinary collaboration opportunities that have shaped the field. This unique approach allows us to reveal previously overlooked patterns and gaps in the literature, offering insights into how research priorities have been set by the global research communities. Our findings contribute to the development of more effective and systemic drought resilience frameworks by quantifying the connections between diverse research topics, ultimately guiding more strategic alignment of efforts among scientists, funding bodies, and policymakers. "

**2 Methods**

Line 60: are they all scientific publications (the 131,748 abstracts)?

Yes, as stated by the source.

Lines 73-74: while describing the "three Tiers" here for the first time, it would be helpful to refer to Fig. 2 already for a clearer schematic representation. This visual aid will enhance understanding and provide an easier reference for readers.

We added a reference to Fig. 2:
"We then selected five topics as a reasonable number for the general classification level, twelve topics for a median level of granularity and fifty topics for the finest level granularity (see results Section Figure 2)."

Line 77: I would emphasize already here in the beginning that the similarity between 2 topics is **within a document** otherwise its not clear until the reader reads about it on Line 79.

Thank you. We clarified accordingly:

"For topic congruence, we calculated the cosine similarity between topic pairs within each individual document."

**3 Results**

**3.1 Major and specific topics in drought research**

Figure 1: I think this Figure needs a clearer explanation in the caption, stating what exactly is shown on the right x axis and the left x axis (especially on the right – the similarity index corresponds to the degree of intedisciplinarity).

We agree and added the degree of interdisciplinarity to the y axis label, which makes it clearer. We also modified the caption to better explain the figure.

[Figure]

"Figure 1: Publications by year in drought research. Research abstracts listed in Scopus and analyzed over the past century with regard to interdisciplinarity. Drought research exhibits an exponential trend (R2 = 0.98). This trend is highlighted by the increasing ratio of drought research to overall research publications. Interdisciplinarity is calculated as cosine similarity index which is the normalized cross-topic intersection within a document. Focus on specific topics increased until 1980s which is marked by a decline in interdisciplinarity. 1980 onwards plant genetics took a rise, leading to ups and downs in interdisciplinarity. From 2007 onwards inter-disciplinary rose again consistently. "

Figure 2: the figure would benefit from adding labels such as "major topics" next to Tier 1 with an indication of "5 topics," "intermediate topics" next to Tier 2 with "12 topics," and "highly specialized topics" next to Tier 3 with "50 topics." This would help readers better orient themselves within the manuscript.

Thank you. We agree and took up the suggestion with the new figure as follows:

[Figure]

Figure 2: the categorization of topics (M,M,I,M,E) lacks clarity, particularly the selection of "E" (events and historical analysis) for the topic "Precipitation and Drought types." Why wouldn't "Methods and processes" be a fitting category for this topic?

We agree that both categories fit well. We reassigned to methods and processes as well as events and historical analysis.

Lines 117 and 119: require consistency in numerical representation, whether as spelled-out words ("twelve topic") or numerals ("12-topic").

Thank you. We now spelled out the numbers.

Figure 3: a question regarding the sequence of the 12 topics displayed in the box: should they align with the sequence seen in Tier 2 of Figure 2? Additionally, it would enhance clarity to label the box containing the 12 topics as "Intermediate Topics Tier 2" at the top, facilitating a quicker understanding of its connection to the preceding Figure 2.

Topic sequence should not align with the sequence in Tier 2.

We added a heading to the legend to facilitate the interpretation of the Figure:

[Figure]

Lines 144-146: quite hard to read this sentence, I would recommend rephrasing it.

We agree. We rephrased the sentence and the paragraph, to make the whole paragraph more clear:

"Following this categorization, tier 3 topics with a strong link to forecasting can be grouped for example into methods related topics (e.g. remote sensing, hydrologic modelling, meteorological drought) and events related topics (e.g. historic drought records and chronology, monsoon (Figure 2))."

**3.2 General and emerging trends**

Line 168: would recommend to add ref to Figure 4 here already? And potentially to explain why to focus only on last 4 decades?

Agreed. We added the reference and a sentence to clarify why four decades:
"We explored the development over time for drought-related research topics and their relative contributions over the past four decades (Figure 4) and more recently, referring to the years 2012-2022. We chose the last four decades because the data showed a rather high variation in relative contributions for the year before 1982."

Line 178-179: the authors should provide references to support this statement. Additionally, it's crucial to note that while increased drought resilience of crops through genetics is valuable, relying solely on this aspect may not suffice, and other mitigation strategies are equally important.

We agree and took this point up in the discussion section:

"Nevertheless, increasing drought crop's drought resilience and forecasting will not suffice as sole mechanisms. Other mitigation strategies at political and ecosystem level can provide equally important and case specific solutions. Knowledge reviews are one tool and required to bridge interdisciplinarity which they already do as illustrated by reviews leading the overall interdisciplinarity score (Figure 6). "

Figure 4: The color choice for "Plant genetics" seems to have changed and become more orange as compared to more yellow assigned to it in Figure 2.

Indeed. We reassigned the color in Figure 4 now:

[Figure]

 Lines 191-194: one of the contributing factors is also the technological advancements that have strengthened forecasting methods.

Thank you. We now added this factor and moved the point to the discussion section (previously in results):
"Shifts in research priorities are influenced by societal interest and advancements in technological capabilities"

**3.3 Interdisciplinarity of drought research**

Lines 208-209: "manifested in sedimentary records and tree ring records" is it referring to "Forest and Fire topic? Would recommend to mention it in the text.

We clarified: "Also, climate records and seasonality are strongly manifested in sedimentary records and tree ring records, causing a high similarity with forests and fire."

Line 236: what exactly does "regional studies" refer to? Isn't it more about spatial characteristics rather than a specific topic?

In fact, it is many different topics with limited geographic context.

Figure 6: I'm surprised to find "Drought Palmer Index" listed as a distinct topic rather than under a broader category like "Drought Index." Additionally, I couldn't find anything related to Forecasting among these 50 topics. Could it be that certain topics, such as Forecasting, have been split into smaller segments (e.g. modelling)?

Yes, this is the case. We clarified the caption of this figure:
"Figure 6: Thematic overlap of the fifty highly specialized research topics. "

Also note, the major general topic forecasting has been split into smaller topics. Forecasting as the more general topic is made up of many smaller topics which are also inter-linked. Modelling and the two drought indices are three out of the fifty more focused sub-topics.

**3.4 Geographic patterns and priorities**

Figure 7: Is it built on 40 or 100 years review, need to mention this. Additionally, wasn't "forecasting" previously indicated in blue but now appears in yellow? Consistency in color usage throughout the manuscript is recommended. Regarding the figure caption, isn't "forecasting" positioned at the top everywhere, not only in Europe and Asia (same as mentioned in the abstract).

Fully agreed. We harmonized the color code for the 12 topics and clarified now that forecasting is positioned at the top i.e. of major interest everywhere.

"Forecasting is everywhere the topic with strongest weight, in particular in Europe and Asia."

[Figure]

**4. Discussion and Future Directions**

**4.1 Implications for research, policy and institutions**

Line 288: its not clear which indicators are meant here? Drought indicators or agricultural impact indicators?

We clarified:
"agricultural impacts and drought indicators"

Lines 288-290: what about the priorities of research community as well and their influence on the trend?

This research community must be very large. An individual's H-Index of 50 can be considered high while on average the 50 sub-topics possess 2620 publications. Looking at the number of publications (>131k) and the potential size of research communities, individual research communities must be of considerable size in order to play a role in the formation of topics.

Line 291: to advance drought impact forecasting, it's crucial to consider several related topics besides genetics.

Please note, while many related topics are considered already, plant traits, genetic breeds and advances seem to be considered rather limited.

We added on several related topics:
"Nevertheless, increasing drought crop's drought resilience and forecasting will not suffice as sole mechanisms. Other mitigation strategies at political and ecosystem level can provide equally important and case specific solutions. Knowledge reviews are one tool and required to bridge

interdisciplinarity which they already do as illustrated by reviews leading the overall interdisciplinarity score (Figure 6)."

Line 298: "knowledge reviews **as** are required …" spelling

This was corrected.

Line 303: here written as "drought indices" but its only "Palmer Drought Index?" within the 50 topics?

We clarified:
"Our results identify flash droughts, the Meteorological Drought Index, and the Palmer Drought Index as distinct, highly specialized topics within the broader drought research literature (see e.g. highly specialized topics in Figure 2). "

**Supplementary**

Couldn't find Figure A and Figure B (Lines 187 and 190) in the Supplementary, I only see Figures S1, S2, S3, S4

This is correct. We corrected the cross-references in Section 2.

**Reply to Reviewer #2**

Black text: Reviewer #2.

Red text: Response by the authors.

The article describes an analysis of bibliographic information (abstracts, keywords, title, publication year, language) performed for the period 1901-2022 to identify trends and focus areas in drought research. In general, the article is well written, contains high quality figures for illustration and addresses an interesting topic. I find in particular the consideration of trends and regional priorities in drought research very interesting. However, I have a number of concerns and suggestions that may be considered to improve the manuscript:

Thank you for the positive overall evaluation of the manuscript. We appreciate your time and effort in reviewing the manuscript. In the forthcoming revision we will consider each of your suggestions.

**General comments:**

The methods applied by the authors are based on a number of important assumptions that should be made transparent to the readers and also being discussed. First of all, the authors limit their analysis to abstract and keywords. This has the advantage that a large number of articles can be considered but the disadvantage, that it is not clear at all how well the specific abstracts and keywords really reflect the content of the articles and the research results.

We addressed and stated this limitation in the discussion. The limitation of only looking at the abstracts and not at the full paper is well noted. While a dependency, the methods and results rely on and benefit from the peer review process carried out on abstracts.

We added:
"4.5    Limitation of Latent Dirichlet Allocation

The data for this research was limited to abstracts, titles, and keywords. While this approach allowed for a large volume of articles to be analyzed over an extended temporal period, it comes with certain limitations. The primary advantage is that abstracts, titles, and keywords are well standardized and harmonized, enabling the inclusion of a vast number of articles. However, a key disadvantage is the uncertainty regarding how accurately these abstracts and keywords reflect the full content and findings of the articles. The results are heavily dependent on the effectiveness of the peer-review process, underscoring the critical role of abstracts in research communication.

A more in-depth study, potentially utilizing more advanced models or normalization procedures, would require additional computational resources (e.g. compare (Callaghan et al., 2020; Ogunleye et al., 2023). For instance, normalizing word frequency could help account for differences in document length. It's important to note that a full-text analysis might reveal that specialized research is more interdisciplinary than it appears from abstract-only analysis. Additionally, the use of semantic analysis could extract further insights and generate more detailed information (Geeganage et al., 2024; Niu et al., 2022)."

Second, it assumes, that terms and definitions used in drought research have been similar across time, regions, disciplines and scales.

This is correct to some degree. We will address this point in the discussion section. The strength of Latent Dirichlet Allocation (LDA) is that it uses all words within a document for analysis. While we

cannot exclude terminology to play a role in topic formation, it is always a number of keywords rather than few which form the context and a single topic.

We added in the discussion section:
"4.1     Definition and use of drought in literature

An important aspect of analyzing the results is understanding the definition, mention, and meaning of the word drought. The definition of drought has been widely discussed in literature. The discussions cover quantitative aspects, such as different drought indicators (Satoh et al., 2021), specific drought events such as flash droughts (Schwartz et al., 2023), and a more generalized concept of drought as driven by and as threat to societal systems (Mishra and Singh, 2010; van Loon et al., 2016b). In line with these discussions, our results identify flash droughts, the Meteorological Drought Index, and the Palmer Drought Index as distinct, highly specialized topics within the broader drought research literature (see e.g. highly specialized topics in Figure 2). At a more general level, drought is perceived as systemic, encompassing ecosystem, societal, and economic dimensions. At this level, case studies are particularly useful for quantifying connection strengths and impacts within specific environments (van Loon et al., 2019; van Loon et al., 2024). In genetic and plant physiological research, drought is defined as a system state that hampers plant growth (Gaudin et al., 2013; Moran et al., 2017). In plant genetics, highly specialized topics focus on methods to foster drought tolerance and drought resistance of plants through e.g. enhancing water use efficiency.

In agreement with literature, this analysis' results and the topological maps of drought research indicate, that there is not a unique understanding of drought. Rather, the different understandings, scale and concepts are interconnected at both general and specific levels. The results build on the strength of LDA as a method, to calculate connection strengths based on co-occurrence and frequency of multiple keywords and topics rather than focussing on specific terms and meanings. Hence, this method results in maps of connection strengths between different systems, foci and perspectives (Figure 5 and Figure 6)."

Certainly, these are strong assumptions. To my experience it is frequently not clear whether statements in the abstract section represent the present understanding in the research community, opinions of the authors, or really conclusions drawn from the research results that are justified by real data. One example is the abstract of the present manuscript. In the last sentence the authors state: "In future, we recommend research and funding agencies to strengthen the track of more interdisciplinary and systemic cross-topic drought research in order to cope with drought as a multi-sectoral risk requiring multi-sectoral response frameworks." Is this recommendation justified by the results of the present analysis? Certainly not. Instead it is the opinion of the authors and reflects well a popular believing of a large part of the drought research community. Nevertheless, this statement can also be challenged. Interdisciplinary, systemic, cross-topic and multi-sectoral research is more expensive and requires more effort than targeted disciplinary research. Funding agencies may have to decide therefore in practice whether they fund 5 small disciplinary projects or one big interdisciplinary project. It seems therefore, that such a general statement is not appropriate.

Certainly, to provide solutions for very specific problems (that nevertheless can have a big impact) such as to improve drought tolerance of crops, to develop more efficient water use technologies or to develop new approaches to improve groundwater recharge, disciplinary research will likely be sufficient. In contrast, to improve drought management in large basins, a multidisciplinary approach may be appropriate. This example shows that the approach to simply extract words and terms from the abstract without considering the other sections and the context of the article may result in misleading conclusions.

First, we agree with reviewer #2 that some abstracts in the corpus may not reflect the content of the paper or may also include misleading statements. This is a qualitative aspect of the source data which sets constrains to the use of literature and is beyond the scope of our analysis.

We added this constraint as outlined above.

Second, we revised the statement(s) in this study's abstract:

"Drought research addresses one of the major natural hazards that threatens progress toward the Sustainable Development Goals (SDGs). This study aims to map the evolution and interdisciplinarity of drought research over time and across regions, offering insights for decision-makers, researchers, and funding agencies. By analyzing more than 130,000 peer-reviewed articles indexed in Scopus from 1901 to 2022 using Latent Dirichlet Allocation (LDA) for topic modeling, we identified distinct shifts in research priorities and emerging trends. The results reveal that plant genetic research for drought-tolerant genotypes and advancements in drought forecasting are the most dominant and continuously growing areas of focus. In contrast, the relative importance of topics such as ecology, water resource management, and forest research has decreased. Geospatial patterns highlight a universal focus on forecasting methods, with a strong secondary emphasis on policy and societal issues in Africa and Oceania. Interdisciplinarity in drought research experienced a marked decline until 1983, followed by a steady increase from 2007 onward, suggesting a growing integration of diverse fields. Emerging topics in recent years signal evolving priorities for future research. This analysis provides a comprehensive overview of drought research trends across sectors and regions, offering strategic guidance for aligning research efforts with drought resilience goals. The findings are crucial for research funding agencies and policymakers aiming to prioritize areas with the highest potential to mitigate drought impacts effectively."

One interesting result of the present study is that the majority of drought related articles address either plant genetics and physiology or drought forecasting. These domains address completely different scales and, even more relevant, may use different definitions what a drought is. At large spatial scales, the terminology is distinguishing drought as an extreme event from aridity as a climate indicator. At plant level drought stress is usually when the water supply is lower than needed for optimal growth. Here, no difference is made between water deficits caused by the fact that crops grow in arid regions and water deficits caused by droughts. Consequently, drought has a different meaning and drought research has another context at these two scales. 3 out of the 12 topics presented in figure 3 (plant stress response, crop breeding, plant genetics) are in a context where drought is understood as a water deficit in general, 7 topics are in contexts that consider drought to be an extreme event and 2 topics (farming, water use efficiency) may use one or the other understanding, depending on the context. While it is as it is, this example shows that the use and meaning of terms may be context specific, a difference that likely cannot be detected by a simple analysis of article abstracts.

I also think that the use of terms in the literature has changed over time. For example, most of the large body of historical articles related to dryland research will have drought in the list of keywords although, according to the present understanding, this research is more linked to aridity and therefore not to drought as an extreme event. Consequently, the results of the trend analysis performed by the authors also have to be taken with care.

Thank you for pointing this out. This is an important finding. We appreciate reviewer #2's analysis and will expand on the definition and meaning of drought in different contexts. We will investigate and note in the discussion, whether the observation that drought can be defined as water deficit in general (plant physiology related) and as an extreme event (forecasting related), and if these two

definitions are overlapping. This related to the notion that the use of drought may have changed over time.

We added this discussion point as outlined above.

**Specific comments:**

**Abstract:** see general comment 1

**Introduction:** is very general and mainly describes why drought research itself is relevant. Research questions are not mentioned and the reader will not be prepared for the analysis described later. There is hardly any link between the introduction and the methods section. Consequently, the reader does not know what the authors what to find out and why they use the specific methods described in the Methods-section. The specific gap of knowledge is not described so that it is difficult for the readers to see the novelty. There are so many review papers in the literature about drought research, please make more clear what the innovation of the present article is and what you make differently compared to the review papers that have been published before.

Thank you. We agree and will strengthened the link between introduction and methods.

"This study is motivated by the need to enhance our understanding of the evolving landscape of drought research, particularly in light of the escalating challenges posed by climate change and water scarcity. While previous reviews outlined the need to synthesize the immense body of literature on drought research (Stein et al., 2022), our analysis distinguishes itself through the use of a data-driven, unsupervised machine learning approach to examine over 130,000 peer-reviewed articles. By exploring long-term research trends, we identify critical shifts in thematic focus, fundamental and emerging trends, and interdisciplinary collaboration opportunities that have shaped the field. This unique approach allows us to reveal previously overlooked patterns and gaps in the literature, offering insights into how research priorities have been set by the global research communities. Our findings contribute to the development of more effective and systemic drought resilience frameworks by quantifying the connections between diverse research topics, ultimately guiding more strategic alignment of efforts among scientists, funding bodies, and policymakers."

**Methods:** It is well described which methods have been used but not why. What is the advantage of using LDA in that specific case, compared to other alternatives?

We now address this point by adding:
"Compared to other alternatives, LDA allows for multiple topics within a single document. Also, LDA represents a compromise between computationally more expensive and more costly topic modelling approachs such as BERTopic (Ogunleye et al., 2023), and simpler and computationally less expensive approaches such as Latent Semantic Analysis (Deerwester et al., 1990)."

**Results:** The first four lines are a summary of the methods but not results.

Thank you. We agree and removed the first four lines of the results section.

Figure 1: Many of the readers will not know what cosine similarity means. Please explain that in the figure caption. That the number of drought related articles has increased over time is not surprising since there are more researchers who write more articles per year, compared to former times. I would find it more interesting to see the change in the relative share in drought related articles in the whole scientific literature.

We added a brief explaination on cosine similarity. We also added "interdisciplinarity" to the axis label.

We added to and modified the caption as follows:
"Interdisciplinarity is calculated as cosine similarity index which is the normalized cross-topic intersection within a document. Focus on specific topics increased until 1980s which is marked by a decline in interdisciplinarity. 1980 onwards plant genetics took a rise, leading to ups and downs in interdisciplinarity. From 2007 onwards inter-disciplinary rose again consistently."

We also added the ratio of drought research to all publications to the figure:

[Figure]

This is also addressed in the results section as follows:
"The proportion of articles focussing on drought increases year by year compared to the general scientific literature. This is expressed by the ratio of drought related research compared to the available scientific publications in webofscience (Figure 1)."

**Discussion:** Please add one section describing the limitations of data and methods used in the present research.

We took up this point.

Details on limitations of data and methods where added as outlined above:

"4.5     Limitation of Latent Dirichlet Allocation

[...]"

**References**

Deerwester, S., Dumais, S. T., Furnas, G. W., Landauer, T. K., and Harshman, R.: Indexing by latent semantic analysis, J. Am. Soc. Inf. Sci., 41, 391–407, https://doi.org/10.1002/(SICI)1097-4571(199009)41:6%3C391:AID-ASI1%3E3.0.CO;2-9, 1990.

Ogunleye, B., Maswera, T., Hirsch, L., Gaudoin, J., and Brunsdon, T.: Comparison of Topic Modelling Approaches in the Banking Context, Applied Sciences, 13, 797, https://doi.org/10.3390/app13020797, 2023.

**Reply to Reviewer #3**

Black text: Reviewer #3.

Red text: Response by the authors.

The authors use a well-known method to give an overview of drought research topics and their development over time. While the method is not new, I find the results and the added thoughts by the authors to be of interest to the research community.

In order to make a topic model research overview interesting and relevant the analysis needs to go beyond reporting and interpret the conclusions for the research community. The authors do this very well. For example, I really like the discussion on including plant genetics into future yield prediction in Section 3.2. My only criticism would be where these interesting discussions currently happen. At the moment these very relevant findings and interpretations are hiding at the end of the individual results sections and are only briefly picked up in the discussion section. I would recommend a clearer split between results and discussion, meaning that e.g. Lines 191 to 196 should be part of the discussion, not the results. Same with other paragraphs. E.g. the discussion around missing topics in L156 to 164 then repeats in L301 to 306.

Thank you for your positive evaluation of the manuscript. We appreciate your time and effort in reviewing the manuscript.

We agree that the paper will benefit from restructuring. We henceforth moved discussion points from the results section to the discussion section.

The discussion section now includes following sections:
"4.1     Definition and use of drought in literature

4.2     Forecasting Methods

4.3     Bridging the Arch of Drought Research

4.4     Implications for research, policy and institutions

4.5     Limitation of Latent Dirichlet Allocation

"

The discussion section on drought resilience frameworks makes an important point (L308). But at the moment it feels more like an afterthought. It could, for example, already be introduced in the introduction as a motivation for a drought research overview and then elaborated more in the discussion.

Thank you. We now integrated drought resilience frameworks already in the introduction:

"Our findings contribute to the development of more effective and systemic drought resilience frameworks by quantifying the connections between diverse research topics, ultimately guiding more strategic alignment of efforts among scientists, funding bodies, and policymakers."

Minor points:

L18: Please already mention that the generative model you are using is LDA, e.g. "using a generative model (topic model)", as for some people that term will be more familiar.

We modified accordingly.

Introduction: Please expand on the motivation for this study. The first three paragraphs repeat well-known information that summarises why drought is a relevant topic. This could be shortened into one paragraph. And then the introduction should introduce the motivation for this specific analysis. Why do we need an analysis of drought research topics? (relevant for drought resilience frameworks?). What is the problem with past drought reviews? (Too many publications to humanly read? (Stein et al, 2022)). There are a lot of bad scientometric reviews out there. What makes this one different?

We will expanded on the motivation for this analysis thereby addressing the questions pointed out: "This study is motivated by the need to enhance our understanding of the evolving landscape of drought research, particularly in light of the escalating challenges posed by climate change and water scarcity. While previous reviews outlined the need to synthesize the immense body of literature on drought research (Stein et al., 2022), our analysis distinguishes itself through the use of a data-driven, unsupervised machine learning approach to examine over 130,000 peer-reviewed articles. By exploring long-term research trends, we identify critical shifts in thematic focus, fundamental and emerging trends, and interdisciplinary collaboration opportunities that have shaped the field. This unique approach allows us to reveal previously overlooked patterns and gaps in the literature, offering insights into how research priorities have been set by the global research communities. Our findings contribute to the development of more effective and systemic drought resilience frameworks by quantifying the connections between diverse research topics, ultimately guiding more strategic alignment of efforts among scientists, funding bodies, and policymakers."

L64: Scopus is not the largest database (see https://help.openalex.org/coverage for a comparison).

Thank you. We modified and added:
"Scopus provides the a curated database of scientific literature and grants access to data- and text mining to licensed users for academic purpose. The following alternative large databases for meta-information of scientific literature were considered: OpenAlex, Web of Science, Dimensions and Semantic Scholars. We chose Scopus because of its high quality of information and granted access for research purpose."

L70: "LDA did cluster" is a strange grammar. Consider rephrasing.

We rephrase to:

"We then calculated topic distributions with LDA for the documents and for given number of topics based on overall and document specific keyword distributions."

L96: The description of the methods repeats here. Consider shortening.

Thank you. We agree and removed the repitition.

L129: Can you add a range? How many papers did the smallest/largest topic include?

We added the range:

"For fifty topics, shares of topics ranged between 134 documents and 5901 documents with a median of 2337 documents. "

L133/140: This might be an editing problem, but there should not be a paragraph break between these two, as the thought continues. In the next line.

We will moved the positioning of the Figure to fit the flow of content.

L152: It would be good to see which topics the author classifies as impact, event and methods related. I would this categorisation also helpful in structuring Figure 6. However, I would understand if a clear categorisation for some topics is difficult.

We now point stronger to figure 2, where classifications are noted. Also we note following sentence, which agrees with the reviewers statement:
"This interplay, however, sometimes leads to challenges in distinctly categorizing topics, as evidenced by occasional overlaps and blurred boundaries among these categories. "

Since classifications are annotated in Figure 2, we restrain from adding them as further content to the already dense Figure 6.

L161: A keyword search across the articles might give an indication which of the two explanations are true.

After consideration of your suggestion, the keyword search did not yield conclusive results. We therefore leave two possible explanations for the reader.

Section 3.2. I think the labels for Figures A and B have been switched for the entire section.

Yes, this is correct. We switched cross-references to Figures A and B accordingly.

Figure 5/6. The upper triangle of the plot only mirrors the lower triangle. Consider removing.

This is correct. We considered removing. However, it makes the figure more accessible, if the labels are sorted by overall similarity and scores can be read line by line, with full cross-reference to the second axis labels. Hence, we suggest to keep both triangles.

L213: Can you elaborate on the surprising geomorphology and water use efficiency connection? In what kind of research do they overlap?

We added a clarification that these two topics have highest overall interdisciplinarity:

"…possess highest overall interdisciplinarity…"

We also added a sentence flushing out this example:

"For example, research where geomorphology and water use efficiency well overlap focuss on soil processes, soil formation and impacts on plant water uptake as well as irrigation."

For your interest: This is based on the following 10 publications (titles and year only), which exhibit highest overlap between these two topics:

- Shoot and root responses of trifolium vesiculosum to boron fertilization in an acidic Brazilian soil (2007).
- Ash and biochar amendment of coarse sandy soil for growing crops under drought conditions (2022).
- Water relations and root activities of Buchloe dactyloides and Zoysia japonica in response to localized soil drying (1999).
- Effect of drought and irrigation on the fate of nitrogen applied to cut permanent grass swards in lysimeters: Leaching losses (1984).
- Photosynthesis, respiration, and carbon allocation of two cool-season perennial grasses in response to surface soil drying (2000).
- Grazing effects on shoot and root dynamics and above and belowground nonstructural carbohydrate in Caucasian bluestem (1988).
- Effects of tillage and seasonal variation of rainfall on soil water content and root growth distribution of winter wheat under rainfed conditions of the Loess Plateau, China (2022).
- Use of water by six grass species: 2. Root distribution and use of soil water (1979).
- Bermudagrass growth in soil supplemented with inorganic amendments (2003).
- Characteristics of root distribution for three kinds of plants under different irrigation treatments in drift desert (2012).

L235: This is the only discussion I did not really like. It sounds a little bit like you are advocating for interdisciplinarity for interdisciplinarity' sake, which I do not agree with. Plus you are only looking at abstracts. A full-text analysis will likely reveal the specialised research to be more interdisciplinary as well.

We agree and actually removed this point.

We noted as well that a full-text analysis will likely reveal stronger overlaps:

"It's important to note that a full-text analysis might reveal that specialized research is more interdisciplinary than it appears from abstract-only analysis. Additionally, the use of semantic analysis could extract further insights and generate more detailed information (Geeganage et al., 2024; Niu et al., 2022)."

Section 3.4. The low numbers of drought research in North America are surprising and need to be checked. The US is always a major contributor in previous research distribution analyses (e.g. Emmer (2018) and Callaghan et al, 2021). Particularly for drought research, I would expect a lot of research with the megadrought in California in recent years.

This is surprising, indeed. One reason for the expectation of high North America output in drought research is the volume of research funding in North America. In contrast to Emmer 2018, we did not attribute geographic location based on (co-)authorship affiliation. We have chosen a similar approach as Callaghan et al. (2021) by using geographic information i.e. country names and for USA federal state names to assign geographic context. Including "usa" in the names leads to 13.5 instead of 10.8 percent, but often there are remnants from text trimming left which is why we prefer to not use "usa" as tag for united states.

L306: Is it possible that the diversity of terms in use for these topics might prevent a clear topic (e.g. compound vs multi-hazard vs hazard cascade…)?

The strength of LDA is that it uses all words within a document for analysis. While we cannot exclude terminology to play a role in topic formation, it is always a number of keywords rather than only a few keywords. These together form the context and topics.

We therefore suggest to not list this point and only name these two explanations:
"First, these topics may not have emerged as distinct areas within the drought research domain and could be distributed across broader research themes. Second, the volume of research on these topics may be insufficient to form distinct clusters. To make a meaningful impact on drought research, these emerging topics need to gain further momentum through increased publication numbers, research focus and funding mechanisms."

Callaghan, M., Schleussner, C. F., Nath, S., Lejeune, Q., Knutson, T. R., Reichstein, M., ... & Minx, J. C. (2021). Machine-learning-based evidence and attribution mapping of 100,000 climate impact studies. Nature climate change, 11(11), 966-972.

Emmer, A. (2018). Geographies and scientometrics of research on natural hazards. Geosciences, 8(10), 382.

Stein, L., Mukkavilli, S. K., & Wagener, T. (2022). Lifelines for a drowning science-improving findability and synthesis of hydrologic publications. Hydrological Processes, 36(11), e14742.